



# Consistency and structural uncertainty of multi-mission GPS radio occultation records

Andrea K. Steiner[1,2], Florian Ladstädter[1,2], Chi O. Ao[3], Hans Gleisner[4], Shu-Peng Ho[5], Doug Hunt[6], Torsten Schmidt[7], Ulrich Foelsche[2,1], Gottfried Kirchengast[1,2], Ying-Hwa Kuo[6], Kent B. Lauritsen[4], Anthony J. Mannucci[3], Johannes K. Nielsen[4], William Schreiner[6], Marc Schwärz[1,2], Sergey Sokolovskiy[6], Stig Syndergaard[4], Jens Wickert[7,8]

[1] Wegener Center for Climate and Global Change (WEGC), University of Graz, Graz, Austria
[2] Institute for Geophysics, Astrophysics, and Meteorology/Institute of Physics, University of Graz, Graz, Austria
[3] Jet Propulsion Laboratory (JPL), California Institute of Technology, Pasadena, CA, USA
[4] Danish Meteorological Institute (DMI), Copenhagen, Denmark
[5] NESDIS/STAR/SMCD, Center for Weather and Climate Prediction, College Park, MD, USA
[6] COSMIC Project Office, University Corporation for Atmospheric Research (UCAR), Boulder, CO, USA
[7] German Research Centre for Geosciences (GFZ), Potsdam, Germany
[8] Technische Universität Berlin, Berlin, Germany

*Correspondence to:* Andrea K. Steiner (andi.steiner@uni-graz.at)

**Abstract.**

Atmospheric climate monitoring requires observations of high-quality conforming to the criteria of the Global Climate Observing System (GCOS). Radio occultation (RO) data based on Global Positioning System (GPS) signals are available since 2001 from several satellite missions with global coverage, high accuracy, and high vertical resolution in the troposphere and lower stratosphere. We assess the consistency and long-term stability of multi-satellite RO observations for use as climate data records. As a measure of long-term stability, we quantify the structural uncertainty of RO data products arising from different processing schemes. We analyze atmospheric variables from bending angle to temperature for four RO missions, CHAMP, Formosat-3/COSMIC, GRACE, and Metop, provided by five data centers. The comparisons are based on profile-to-profile differences, aggregated to monthly means. Structural uncertainty in trends is found lowest from 8 km to 25 km altitude globally for all inspected RO variables and missions. For temperature, it is <0.05 K per decade in the global mean and <0.1 K per decade at all latitudes. Above 25 km, the uncertainty increases for CHAMP while data from the other missions are based on advanced receivers and are usable to higher altitudes for climate trend studies: dry temperature to 35 km, refractivity to 40 km, and bending angle to 50 km. Larger differences in RO data at high altitudes and latitudes are mainly due to different implementation choices in the retrievals. The intercomparison helped to further enhance the maturity of the RO record and confirms the climate quality of multi-satellite RO observations towards establishing a GCOS climate data record.



# 1 Introduction

Consistent and long-term stable observations are critically important for monitoring the Earth's changing climate. In the free atmosphere above the boundary layer, uncertainties across data sets can be substantial and observations of thermodynamic variables are sparse, especially when considering measurements capable of detecting changes in the climate state. This was

identified as a key issue in the Fifth Assessment report of the Intergovernmental Panel on Climate Change (IPCC), stating the need for data with better accuracy for monitoring and detecting atmospheric climate change, particularly in the upper troposphere and in the stratosphere (Hartmann et al., 2013).

In order to ensure global homogenous and accurate measurements, the Global Climate Observing System (GCOS) Program defined basic monitoring principles for climate data generation (GCOS, 2010 a; b), and requirements for Climate Data Records

(CDRs) of Essential Climate Variables (ECVs), such as air temperature (GCOS, 2016). A CDR is based on a series of instruments with sufficient calibration and quality control for the generation of homogeneous products. This means that separate data sets from different platforms must be directly comparable to give reliable long-term records, accurate and stable enough for climate monitoring (GCOS, 2010 a), which requires that the observations are traceable to standards of the international system of units (SI) (Ohring, 2007).

For climate observations, the accuracy requirement is much more stringent than for weather observations, e.g., 0.1 K versus 1 K for temperature (Trenberth et al., 2013). However, the key attribute is long-term stability. The uncertainty must be smaller than the signal expected for decadal change (Ohring et al., 2005; Bojinski et al., 2014). Accordingly, ECV product requirements for air temperature include global coverage, a vertical resolution of 1–2 km in the troposphere and the stratosphere, a horizontal resolution of 100 km, a measurement uncertainty of 0.5 K, and a stability of 0.05 K per decade (GCOS, 2016).

Global Navigation Satellite System (GNSS) radio occultation (RO) has been identified as a key component for the GCOS due to its potential of being a climate benchmark record (GCOS, 2011). Efforts of the RO community are ongoing since the pioneering GPS/MET proof-of-concept mission in 1995 (Ware et al. 1996; Kursinski et al. 1997; Rocken et al. 1997; Steiner et al. 1999; 2001) to establish GNSS RO as observing system for Earth's atmosphere and climate. Since 2001, continuous observations are available from several RO satellite missions with beneficial properties for climate use. Most missions have

used only GPS signals so far, including the ones analyzed in this study; multi-GNSS use started with the Chinese FY-3C RO mission that also exploits Beidou System (BDS) signals (Bai et al., 2018; Sun et al,, 2018).

RO is a limb sounding technique based on GNSS radio signals, which are refracted and retarded by the atmospheric refractivity field during their propagation to a receiver on a Low Earth Orbit (LEO) satellite. An occultation event occurs when a GNSS satellite sets behind (or rises from behind) the horizon. Its signals are then occulted by the Earth's limb from the viewpoint of

the receiver. The atmosphere is scanned vertically through the relative movements of the satellites providing a good vertical resolution. RO accurately measures the Doppler shifts of the refracted signals by relying on precise atomic clocks, which enables traceability to the SI unit of the second (Leroy et al., 2006), long-term stability, and small uncertainties. Therefore, a seamless observation record can be formed using data from different missions without the need for inter-calibration nor





temporal overlap (Foelsche et al., 2011; Angerer et al., 2017). Observations are available in nearly all-weather conditions as signals in the L-band microwave range are not affected by clouds.

GNSS RO provides high vertical resolution profiles of atmospheric bending angle and refractive index that relate directly to temperature under dry atmospheric conditions, where water vapor influence is negligible. For moist atmospheric conditions, in the troposphere, a priori information is needed in the retrieval. The vertical resolution is typically about 100 m in the lower troposphere to about 1 km in the stratosphere (Kursinski et al. 1997; Gorbunov et al. 2004). Zeng et al. (2019) established vertical resolution as 100–200 m near the tropopause, about 500 m in the lower stratosphere at low to mid-latitudes, and about 1.4 km at 22–27 km at high latitudes.

Data products comprise profiles and gridded fields of bending angle, refractivity, pressure, geopotential height, temperature, and specific humidity, for use in atmosphere and climate studies (see the reviews of Anthes et al., 2011; Steiner et al., 2011; Ho et al., 2019 in press). Various derived quantities include, e.g., planetary boundary layer height (e.g., Sokolovskiy et al., 2006; Xie et al., 2006; Guo et al., 2011; Ao et al., 2012; Ho et al., 2015), tropopause parameters (e.g., Randel et al., 2003; Schmidt et al., 2005; 2008; Rieckh et al., 2014), and geostrophic wind (e.g., Verkhoglyadova et al., 2014; Scherllin-Pirscher et al., 2014). RO provides atmospheric profiles with essentially independent information on altitude and pressure. This unique property ensures equivalent data quality on different vertical coordinates, i.e., mean-sea-level (MSL) altitude, geopotential height, pressure levels, or potential temperature coordinates (Scherllin-Pirscher et al., 2017).

RO observations improve weather prediction (Healy et al., 2005; Aparicio and Deblonde, 2008; Cardinali, 2009; Cucurull, 2010; Cardinali and Healy, 2014) and hurricane forecasts (e.g., Huang et al., 2005; Kuo et al., 2009; Liu et al. 2012; Chen et al., 2015; Ho et al., 2019). The RO data anchor atmospheric (re)analyses (Poli et al., 2010; Bauer et al., 2014; Simmons et al., 2017), and are useful for validating other types of observations (e.g., Steiner et al., 2007; He et al., 2009; Ladstädter et al., 2011; 2015; Ho et al., 2009a; 2010; 2017; 2018) and climate models (Ao et al., 2015; Pincus et al., 2017; Steiner et al., 2018). The importance of the RO record for climate monitoring grows with its increasing length (e.g., Steiner et al., 2009; Schmidt et al., 2010; Lackner et al., 2011; Steiner et al., 2011; Gleisner et al., 2015; Khaykin et al., 2017; Leroy et al., 2018).

An important prerequisite for CDRs is information on uncertainties of the provided variables. For individual RO temperature profiles, the observational uncertainty estimate is 0.7 K in the tropopause region, slightly decreasing into the troposphere and gradually increasing into the stratosphere (Scherllin-Pirscher et al., 2011a; 2017). For monthly-zonal averaged temperature fields, the total uncertainty estimate is smaller than 0.15 K in the upper troposphere-lower stratosphere (UTLS), and up to 0.6 K at higher latitudes in wintertime (Scherllin-Pirscher et al., 2011b). Overall, the uncertainties of RO climatological fields are small compared to any other UTLS observing system for thermodynamic atmospheric variables. An overview of the main properties of RO is given in Steiner et al. (2011).

The systematic assessment of accuracy and quality of RO records is in the focus of joint studies by the RO Trends intercomparison working group, an international collaboration of RO processing centers since 2006 (http://irowg.org/projects/rotrends/). The aim is to validate RO as a climate benchmark by comparing trends in RO products





determined by different retrieval centers. This is assessed by quantifying the structural uncertainty (SU) in RO products arising from different processing schemes.

In the first intercomparison studies, we so far quantified the structural uncertainty of RO data from the CHAMP mission (CHAllenging Minisatellite Payload for geoscientific research) provided by different RO data centers. Profile-to-profile

intercomparisons (Ho et al., 2009b; 2012) were based on exactly the same set of profiles from each data center. Complementarily, we compared RO gridded climate records, based on the full set of profiles provided by each center and accounted for the different sampling (Steiner et al., 2013a). The results for gridded CHAMP records were consistent with those for individual profiles. The structural uncertainty in the CHAMP RO record was found lowest in the tropics and mid-latitudes at 8–25 km and to increase above and at high latitudes due to different choices in the retrievals.

Here we present an advanced assessment of the consistency of multi-year RO records for multiple satellite missions and for the full set of dry and moist atmospheric variables. We systematically intercompare RO data products provided by five international RO processing centers and we quantify the structural uncertainty for nine RO climate variables from bending angle to temperature and specific humidity. The comparisons are based on profile-to-profile differences, aggregated to monthly means. We discuss the results with respect to GCOS stability requirements for climate variables. The quantification of

structural uncertainty as one property of a climate benchmark data type is regarded as an essential advance towards a multi-year RO climate record.

In this respect, our study contributes to enhancing the maturity of RO data (Bates and Privette, 2012; Merchant et al., 2017), which is a goal of the RO-CLIM project (http://www.scope-cm.org/projects/scm-08/) within the initiative on Sustained and COordinated Processing of Environmental satellite data for Climate Monitoring (SCOPE-CM). SCOPE-CM supports the

coordination of international activities to generate CDRs. It is also a recommendation of the WMO/CGMS International RO Working Group (IROWG, www.irowg.org) to establish RO-based CDRs at the quality standards of the GCOS climate monitoring principles (IROWG, 2018).

In the following, we give a concise description of the RO data sets and the data processing in section 2. In section 3 we describe the study setup and methods. We present and discuss results on the consistency and structural uncertainty of multi-satellite RO

products in section 3. Section 4 closes with a summary and conclusions.

## 2 Radio occultation data and processing description

The first continuous RO measurements were provided by the German mission CHAMP from May 2001 to October 2008, tracking about 250 RO events per day with a BlackJack GPS receiver (Wickert et al., 2004; 2009). The US/German GRACE (Gravity Recovery and Climate Experiment) twin-satellites (GRACE-A and GRACE-B) were launched in 2002 (Wickert et

al., 2005; Beyerle et al., 2005). RO measurements have been provided since 2006, when the BlackJack receivers onboard GRACE were switched on. As the first constellation mission, the Taiwan/U.S. Formosat-3/COSMIC (Constellation Observing System for Meteorology, Ionosphere, and Climate/ Formosa Satellite Mission 3; denoted F3C hereafter) mission consists of



six satellites for RO observations (Anthes et al., 2008). Launched in 2006, the Integrated GPS Occultation Receiver (IGOR) tracked both setting and rising occultations, resulting in about 500 RO events per day. The Metop series (Luntama et al., 2008) is operated by the European Organisation for the Exploitation of Meteorological Satellites (EUMETSAT). Metop-A has delivered data since the end of 2007 and Metop-B since spring 2013; Metop-C only started data delivery early 2019. All three

Metop satellites carry a GNSS receiver for Atmospheric Sounding (GRAS) with four dual-frequency channels for simultaneous tracking of two rising and two setting events, yielding about 700 observed RO events per day.

Data from these four satellite missions have been delivered for the assessment of the consistency of multi-satellite RO records. The following processing centers provided reprocessed RO data products from bending angle to temperature for this study: Danish Meteorological Institute (DMI) Copenhagen, Denmark; German Research Centre for Geosciences (GFZ) Potsdam,

Germany; Jet Propulsion Laboratory (JPL) Pasadena, CA, USA; University Corporation for Atmospheric Research (UCAR) Boulder, CO USA; and Wegener Center/University of Graz (WEGC), Graz, Austria. Each center has implemented an independently developed processing system for the retrieval of RO data products. While the basic steps in the retrieval (Kursinski et al., 1997) are essentially the same, different implementation options are chosen by the centers for specific processing steps. We briefly describe the basic retrieval steps for dry and moist atmospheric conditions. In Table 1, we present

a concise overview on the implementation of the processing steps at each center and discuss the main differences.

The fundamental measurement is the GNSS signal phase change as function of time, which varies according to the optical path length between the transmitter satellite and the LEO receiver satellite. Highly accurate atomic clocks are the heart of the system ensuring long-term frequency stability. Two coherent carrier signals are transmitted, in case of the U.S. Global Positioning System (GPS) at wavelengths of 0.19 m (L1 signal) and 0.24 m (L2 signal), to remove contributions due to Earth's ionosphere

(Hofmann-Wellenhof et al., 2008; Teunissen and Montenbruck, 2017).

In the retrieval, the Doppler shift, i.e., the time-derivative of the phase, is propagated further (e.g. Melbourne et al., 1994; Kursinski et al., 1997). The kinematic contribution to the Doppler shift due to the relative motion of the GNSS and LEO satellites is determined from precise position and velocity information, i.e., precise orbit determination (POD) (Bertiger et al., 1994; König et al., 2006). Removing it yields the Doppler shift due to the Earth's refractivity field. Errors in the receiver clock

are removed by single differencing with a second reference satellite link or with double differencing, by using additional ground clock information (Wickert et al., 2002). No differencing is needed, i.e., zero differencing, if there are ultra-stable clocks aboard the LEO satellites and clock errors are very small, such as for GRACE or Metop (e.g., Wickert et al., 2002; Schreiner et al., 2010; 2011; Bai et al. 2017). Geodetic processing systems are used to estimate errors in the GNSS transmitter clocks.

For microwave refraction, geometric optics is applied to convert Doppler shift to bending angle profiles, assuming local spherical symmetry of the atmosphere. In the lower troposphere, multipath and diffraction effects become important due to atmospheric humidity. Here, wave optics methods are applied for the retrieval of bending angle, using phase and amplitude information (e.g., Gorbunov, 2002; Jensen et al., 2003; 2004; Gorbunov et al., 2004; Sokolovskiy et al., 2007). The ionospheric contribution to the signal is largely removed by differencing the dual-frequency GNSS signals, typically at bending angle level



(Vorob'ev and Krasil'nikova, 1994). The ionosphere-corrected bending angle represents the cumulative signal refraction due to atmospheric density gradients.

The next retrieval step is the computation of refractivity from bending angle by an Abel transform (Fjeldbo et al., 1971). This involves an integral with an upper bound of infinity. Also, the signal-to-noise ratio of the bending angle decreases with

increasing altitude (above about 50 km depending on the thermal noise of the receiver). Therefore, an initialization of bending angle profiles with background information is performed at high altitudes. The optimized bending angle profiles are then converted to refractivity profiles.

Refractivity at microwave wavelengths in the neutral atmosphere mainly depends on thermodynamic conditions of the dry and the moist atmosphere, and is given by the Smith Weintraub formula (Smith and Weintraub, 1953). Dry density profiles are

calculated from atmospheric refractivity by neglecting the wet term in the formula. Dry pressure profiles are retrieved using the hydrostatic equation and dry temperature profiles using the equation of state for dry air conditions in the upper troposphere and lower stratosphere. In the lower to middle troposphere, the retrieval of (physical) atmospheric temperature or humidity requires additional background information in order to resolve the wet-dry ambiguity information inherent in refractivity (e.g., Kursinski et al., 1996; Healy and Eyre, 2000; Kursinski and Gebhard, 2014). Different methods are applied for moist air

retrievals including a priori knowledge of the state of the atmosphere. Finally, quality control (QC) is implemented at several processing steps.

Atmospheric profiles are provided as function of mean sea level (MSL) altitude due to accurate knowledge of transmitter and receiver positions (and the assumption of local spherical symmetry), referred to the world geodetic system 1984 (WGS 84) reference coordinate system and the Earth's geoid (see Table 1). The vertical integration of density also provides pressure as

function of altitude. This enables the computation of geopotential height (without the need of information on surface pressure or any other information except gravity potential). Further details on vertical coordinates and geolocation of RO are given in Scherllin-Pirscher et al. (2017).

Table 1 provides an overview on current state-of-the-art retrieval versions and the processing steps implemented at each center as well as information on data description and availability (Steiner et al., 2013; Table 1 updated for current processing versions

and extended for moist air processing steps). Some centers do not start their processing chain directly from the raw data level but start at the level of phase data that has been calibrated to remove geometric effects and clock errors (so-called "excess phase"). DMI and WEGC use calibrated phase and orbit data from UCAR/CDAAC (COSMIC Data Analysis and Archive Center) for all satellite missions in this study. Thus, as three centers start with the same phase and orbit data, the RO products are not independent.

The main differences between the centers' processing steps include the initialization of the Abel integral that transforms bending angles to refractivity, moist-air retrieval, and quality control. For the bending angle vertical profiles, JPL performs an extrapolation of the bending angle to higher altitudes while the other centers apply statistical optimization methods that combine the bending angle measurements with a background bending angle. Each center uses different background information, either atmosphere model climatologies (GFZ, UCAR), observation based climatologies (DMI), or short-range



forecasts (WEGC). Handling of observational and background errors affects the amount of information from observations and from the background included in the retrieved optimized bending angle. Observational error is typically smaller in data from RO systems with improved performance, i.e., lower thermal noise or higher gain antennas enabling higher signal-to-noise-ratio up to higher altitudes.

In the different moist air retrieval implementations, a priori information is also included, stemming either from atmospheric analyses or forecasts (JPL, WEGC) or (model forecast produced with) reanalysis (DMI, UCAR) data. Quality control is handled differently at each center, which means that not always the same set of profiles is delivered by each center.

Figure 1 shows the number of profiles per month delivered by each center for each RO mission. Indicated is also the number of profiles in the common subsets, which we used in the profile-to-profile intercomparison for quantifying structural

uncertainty. For CHAMP, GFZ delivered the largest number of data, followed by UCAR, DMI, WEGC, and JPL. There is a data gap in July 2006 when CHAMP had only very few measurements. The common subset of profiles for CHAMP summed up on average to about 1500 profiles per month.

For F3C, DMI, UCAR, and WEGC delivered nearly the same number of data, only JPL provided a smaller amount. GFZ did not process F3C data. The number of F3C measurements was highest from 2007 to 2010 with more than 70000 profiles per

month and decreased over time as the satellites successively ceased achieving full function. The mission design life was two years. Only two of the six F3C satellites still produced data in 2018. For this study, UCAR provided a reprocessed F3C data set until 04/2014. The common subset of F3C data ranged from 20000 up to 50000 profiles per month over time.

Data for GRACE were provided by three centers, DMI, GFZ, WEGC, delivering nearly the same amount of profiles with a common subset of about 3000 profiles per month. Metop data were provided by DMI, UCAR, and WEGC, with a common

subset of about 15000 profiles increasing to 25000 per month when the second Metop satellite started measuring. The number of common profiles is noticeably less than the number of profiles delivered by any of the centers which is due to the different quality control handling.

## 3 Study setup and method

We investigated the structural uncertainty of the following RO variables: bending angle ($\alpha$), optimized bending angle ($\alpha_{\mathrm{opt}}$),

refractivity ($N$), dry pressure ($p_{\mathrm{dry}}$), dry temperature ($T_{\mathrm{dry}}$), dry geopotential height ($Z_{\mathrm{dry}}$), pressure ($p$), temperature ($T$), and specific humidity ($q$). The atmospheric profiles were provided on a 100 m MSL altitude grid, except bending angle which was given on impact altitude, and geopotential height which was related to dry pressure levels, i.e., "dry pressure altitude" defined as $z_p[\mathrm{m}] = (7000\ \mathrm{m}) \cdot \ln(1013.25\ \mathrm{hPa}/\,p_{\mathrm{dry}}\ [\mathrm{hPa}])$.

Table 2 summarizes the data delivered for this study by each center and gives information on satellite missions, time periods

and atmospheric variables. Not all of the centers provided data for each satellite and each variable. UCAR did not provide optimized bending angle profiles. GFZ did not provide moist-air variables. This was adequately considered in the computations.





The study was based on the intercomparison of collocated profiles between the centers for each satellite mission and atmospheric variable. The profiles were collocated based on a unique event identifier (ID) including information on receiver ID, GPS satellite ID, date, and time of the observation. This means that only the common time periods can be intercompared for which each center provided data continuously. The investigated periods are 09/2001–09/2008 for CHAMP (5 centers),

03/2007–12/2016 for GRACE (3 centers), 08/2006–04/2014 for F3C (4 centers), and 03/2008–12/2015 for Metop (3 centers). We first calculated the differences of each center to the all-center mean (i.e. mean of all centers) over time. By using difference time series we remove the climate variability that is common in the data sets. The remaining deviations that may then be caused by different processing methods. Anomaly difference time series were then computed by subtracting the mean annual cycle for the respective time period (see Table 2) to reduce the natural variability in the differences. Percentage anomaly difference

time series were computed for variables which exponentially decrease with altitude. The spread of linear trends in the anomaly differences and finally the standard deviation of center trends are used for estimating the structural uncertainty of RO records (Wigley, 2006). For each atmospheric variable and satellite mission, we computed the linear trend over the respective time period for the all-center mean, and for each center, as well as the standard deviation of the center trends.

We performed the calculations for each atmospheric parameter ($X$) for each satellite ($s$) of each center ($c$) given at monthly

resolution ($t$) for latitude bands ($\phi$) and altitude levels ($z$), i.e. nine parameters, four satellites, five centers, for 18 latitude bands and up to 600 altitude levels as well as for six large latitude bands and up to 8 altitude layers (after Steiner et al., 2013; Mochart, 2018).

The all-center mean $X_{all}$ of an atmospheric parameter $X$ as function of latitude ($\phi$), altitude ($z$), time ($t$), and satellite ($s$) was calculated using Eq. 1:

$$\overline{X}_{all}\left(\phi_j, z_k, t_l, s_m\right) = \frac{1}{n_{center}} \sum_{i=1}^{n_{center}} X\left(c_i, \phi_j, z_k, t_l, s_m\right). \tag{1}$$

The difference of each center ($c$) to the all-center mean was then calculated for each latitude band, altitude layer, time step, and satellite using Eq. 2, and the percentage difference using Eq. 3:

$$\Delta X\left(c_i, \phi_j, z_k, t_l, s_m\right) = X\left(c_i, \phi_j, z_k, t_l, s_m\right) - \overline{X}_{all}\left(\phi_j, z_k, t_l, s_m\right), \tag{2}$$

$$\frac{\Delta X}{X}\left(c_i, \phi_j, z_k, t_l, s_m\right) = 100 \frac{X\left(c_i, \phi_j, z_k, t_l, s_m\right) - \overline{X}_{all}\left(\phi_j, z_k, t_l, s_m\right)}{\overline{X}_{all}\left(\phi_j, z_k, t_l, s_m\right)}. \tag{3}$$

The mean difference of each center to the all-center mean was computed by averaging over the satellite dependent period, with $n_{time}$ as the number of time steps (months), using Eq. 4:

$$\overline{\Delta X}\left(c_i, \phi_j, z_k, s_m\right) = \frac{1}{n_{time}} \sum_{l=1}^{n_{time}} \left[ X\left(c_i, \phi_j, z_k, t_l, s_m\right) - \overline{X}_{all}\left(\phi_j, z_k, t_l, s_m\right) \right]. \tag{4}$$

Then, the annual cycle for the differences to the all-center mean was computed using Eq. 5. The number of years over which the annual cycle was calculated is denoted $n_{yr}$, the index $l'$ denotes one of the twelve months of a year:



$$\Delta \overline{X}_{AnnCycle}(c_i, \phi_j, z_k, t_{l'}, s_m) = \frac{1}{n_{yr}} \sum_{l''=1}^{n_{yr}} \Delta X(c_i, \phi_j, z_k, t_{l''}, s_m). \tag{5}$$

Subtracting the annual cycle provided the de-seasonalized anomaly differences for each center $c$ and satellite $s$, obtained according to Eq. 6:

$$\Delta X_{DeseasAnomDiff}(c_i, \phi_j, z_k, t_l, s_m) = \Delta X(c_i, \phi_j, z_k, t_l, s_m) - \Delta \overline{X}_{AnnCycle}(c_i, \phi_j, z_k, t_{l\,mod12}, s_m). \tag{6}$$

Fractional (percentage) de-seasonalized anomaly differences were computed analogously.

Linear trends were then computed with standard linear regression for the de-seasonalized anomaly difference time series and, finally, for the de-seasonalized time series of each center. For better comparison, the trends are stated per 10 years. However, we do not discuss trends here as the time periods are different for each RO mission. We are interested in the structural uncertainty of trends represented by the standard deviation of the $n_{center}$ individual center trends. This measure gives us an
indication of the stability of the multi-satellite RO records.

We performed the computations for 10°-zonal means, averaging the collocated individual RO profiles on the given vertical grid on a monthly mean basis. We then averaged to larger latitudinal domains and altitude layers, in which RO data show similar behavior and similar structural uncertainty. We defined six latitude bands: the tropics (TRO; 20°N–20°S), northern/southern mid latitudes (NML/SML; 20°N/S–60°N/S), northern/southern high latitudes (NHL/SHL; 60°N/S–90°N/S),
and a global band (GLOB; 90°N–90°S). We defined (up to) eight altitude layers. The uppermost altitude levels are 60 km for bending angle, 50 km for refractivity, and 40 km for the other variables except humidity (15 km). The inspected vertical layers include 8–18 km, 18–25 km, 25–30 km, 30–35 km, 35–40 km, 40–50 km, 50–60 km. Structural uncertainty in trends is finally presented at the full 100 m altitude grid.

## 4 Results and discussion

**4.1 Comparison of differences in multi-satellite RO profiles for one exemplary month and for the total mean**

For a first overview, we present comparison results for one exemplary month, July 2008, for selected atmospheric RO variables in order to introduce several characteristic features. Figure 2 shows the global mean difference of profiles from each center with respect to the all-center mean, for the missions CHAMP, F3C, GRACE, and Metop. Differences for the variables bending angle, optimized bending angle, refractivity, dry temperature, and physical temperature are presented.
The mean difference profiles for non-optimized bending angle and bending angle are smaller at upper altitudes for F3C, GRACE, and Metop compared to CHAMP due to enhanced receiver quality, and smoother due to the larger number of data available. For CHAMP, the bending angle becomes noisy near 35–40 km and above 40–50 km for the other RO missions. The optimization of the bending angle reduces the noise and stabilizes the retrieval at high altitudes above 50 km. The noise reduction is visible in the optimized bending angle differences, specifically for F3C, GRACE, and Metop. The bending angle
differences are <0.1% from 10–40/50 km impact altitude, depending on the mission.





In the RO retrieval chain of further derived parameters, such as refractivity, pressure, or dry temperature, the impact of background information propagates further downward in altitude for each retrieved parameter. Refractivity, which is proportional to atmospheric density, shows differences of <0.05% at 10–30 km for all satellites in July 2008. Dry temperature differences are small from 8–25/30 km depending on the mission. Physical temperatures, usually derived with a priori

information, show generally smaller differences, with JPL showing larger deviations due to cut-off artefacts below 15 km (see below).

Next, we give an overview on mean differences with respect to the all-center mean, averaged over the full time period of a mission, which we exemplarily show for the F3C mission. Figure 3 presents averaged anomaly differences for bending angle, refractivity, dry temperature, temperature, and specific humidity for 10°-zonal means at a 100 m vertical grid. The mean

differences for bending angle are found to be very small (0.1–0.2%) at all latitudes, except at high latitudes, where differences are larger for JPL and UCAR bending angles. Different choices for the bending angle initialization by the centers are reflected in larger refractivity differences above about 40 km, while below the mean differences are very small (<0.1%). For subsequent derived variables, the differences become larger already above 30 km as seen for dry temperature. There, some latitude dependent features appear which might stem from high-altitude initialization in the retrieval, specifically at high latitudes. At

5–30 km, mean differences for dry temperature are found <0.2 K for all latitude bands. Physical temperature shows similar differences of <0.2 K at 2–30 km altitude. JPL provides physical temperature products only down to a certain altitude. RO temperature is cut off when it rises above 240 K in their moist air retrieval, where background temperature information from ECMWF (European Centre for Medium-Range Weather Forecasts) analyses is used to derive specific humidity. DMI and UCAR use a 1-dimensional variational (1D-Var) method to derive temperature with ERA-Interim (ECMWF Reanalysis-

Interim) products as background. WEGC applies a simplified 1D-Var retrieval method, using ECMWF analyses as background below about 16 km altitude. Above this altitude, WEGC dry and physical temperatures are the same. For specific humidity we find mean differences of each center to the all-center mean of <15%. JPL provides specific humidity data up to 10 km altitude only in synergy with the temperature cut-off, and the number of data decreases above 8 km. The larger differences at this altitude are artefacts and can be removed with a more rigid cut-off. Only few centers delivered humidity and the data have

different height availability, which hampers a rigorous statistical intercomparison of humidity in this study. We thus do not show further comparisons here.

Comparison of mean differences with data of the other satellite missions CHAMP, GRACE, and Metop shows good consistency over the same regions, however, differences are found smaller at higher altitudes, specifically for Metop. Commonalities and differences are further investigated in the full difference time series and revealed in the structural

uncertainty estimates.

## 4.2 Comparison of anomaly difference time series

Here, we investigate anomaly difference time series (see Eq. 6) for each satellite mission (CHAMP, F3C, GRACE, Metop) over the respective time periods as presented in Fig. 4 to 7. We show monthly mean differences to the all-center mean for two





selected variables, bending angle and dry temperature. Bending angle stands at the beginning of the processing chain (after phase data processing) while dry temperature is one of the final RO products commonly used in climate studies. We present results for the global mean (GLO) and for selected zonal means, the tropics (TRO), and high latitudes (SHL, NHL). We do not show results for the mid-latitude bands (NML, SML) as the results are similar to those in the tropics. We investigate

consistencies and deviations in the anomaly difference time series of individual centers to the all-center mean.

For all satellite missions we find that bending angle differences are overall very small and consistent below 30 km at all latitudes. However, there are some differences which we discuss in the following. For CHAMP, the spread of mean anomaly difference trends in bending angle (Fig.4a) is larger than for the other missions. For the zonal means, it is about ±0.05% below 25 km, increasing to about ±0.1% above. At SHL, a larger difference trend is seen for GFZ at 25–30 km. Larger variability in

bending angle is found for JPL over the investigated period, likely due to the sensitivity of the bending angle extrapolation to measurement noise. The difference time series in CHAMP bending angle show similar behavior at high latitudes and in the tropics. The global mean difference trends for CHAMP are -0.03% to 0.06 % at 8–18 km and ±0.02% above.

For COSMIC, the spread of mean anomaly difference trends (Fig. 5a) is found larger at high latitudes than in the tropics. Largest difference trends are found at SHL, with a spread of -0.18% to 0.1% in all altitude layers. This is due to a small shift

in UCAR bending angle in 2013, which is currently under investigation. In the tropics, the differences are small. Only at lower altitudes, variability is larger in the WEGC bending angle. These are artefacts due to improper cut-off of the WEGC profiles below 12 km altitude. However, in the global mean, the spread in difference trends is quite small with -0.02% to 0.1% at 18–25 km, and ±0.01% at 25–30 km, which is smaller than for CHAMP.

GRACE shows highly consistent anomaly differences (Fig. 6a) and a similar behavior at all latitudes. An interesting feature

in GFZ bending angle is an oscillating variability over time for GRACE data. However, the spread in difference trends is very small with ±0.01% in all altitude layers. Globally it is zero. Also, for Metop we find high consistency in anomaly differences (Fig. 7a), with a spread in difference trends of ±0.01% for bending angle except in the tropical band. There, differences are a bit larger with ± 0.05% at 18–25 km.

For refractivity, we find high consistency in the difference trends (not shown here for the time series, but later in section 4.3).

The spread of the difference trends is about ±0.01% to ±0.02% at all latitudes at 8–30 km for F3C, GRACE, and Metop, and near zero globally. For CHAMP, it is within ±0.02% to ±0.03%, and larger differences only occur for GFZ time series at high latitudes.

For dry temperature, the difference time series show some common features for all satellites. We find that the spread in anomaly difference trends for dry temperature is smallest in the troposphere layer (8–18 km), larger in the lower stratosphere

layer (18–25 km), and further increases above. The spread in difference trends is found largest for CHAMP (Fig. 4b), followed by COSMIC (Fig. 5b), GRACE (Fig. 6b), and Metop (Fig. 7b).

The global mean difference trends of the centers for CHAMP ranges from about ±0.08 K at 8–18 km to ±0.16 K at 18–25 km, and reaches -0.31 K to +0.47 K at 25–30 km. For COSMIC, the global spread is only ±0.02 K at 8–25 km to ±0.05 K at 25–



30 km. For GRACE, it is even smaller with ±0.01 K at lower altitudes increasing to ±0.08K at 25–30 km. For Metop, it is near-zero in the troposphere, ±0.01 K in the lower stratosphere, and -0.04 K to +0.02 K above.

For CHAMP dry temperature, some larger differences occur in the tropics. There, the JPL time series show a slight shift, which is most prominent at upper altitude levels. Some deviations occur in the UCAR time series for some winter months at NHL.

These peaks are only visible for a few months, when sudden stratospheric warmings occurred. The peaks can be explained by high altitude initialization with the NCAR climatology, which does not capture the extraordinary large temperature changes at high latitudes during sudden stratospheric warmings. For GRACE, a peak in WEGC data is seen at the beginning of the time series at upper height levels. However, in the global average, the anomaly differences are found very small despite some larger deviations in some NHL winter months. Also, the results for physical temperature are in very good agreement. They are

consistent with dry temperature and UCAR data peaks are reduced to about 50%.

Comparing results of the four RO missions, we find the highest consistency for GRACE and Metop between the centers. CHAMP and COSMIC show a bit larger differences. Apart from small features, the results are very consistent at 8–30 km. One potential reason for the higher consistency of GRACE and Metop records is considered to be technological advances on the newer satellite generations. Partly it might also be due to that only three centers delivered data for these missions, while

five centers provided data for CHAMP and four centers provided data for F3C.

## 4.3 Structural uncertainty for RO multi-satellite records

Finally, we analyzed the consistency of trends for multi-satellite records from five different processing centers. We calculated trends for all variables based on the anomaly time series of the individual centers. We also computed the all-center mean trend. The spread of the center trends, i.e., the standard deviation of the individual center trends, is taken as a measure for the structural

uncertainty of the RO records. We stress at this point that we do not investigate nor interpret climatological trends because this is not the focus of this study. Here, we are interested in the structural uncertainty of the RO records.

We present trends and standard deviations for each RO mission separately, for CHAMP (Fig. 8), F3C (Fig. 9), GRACE (Fig. 10), and Metop (Fig. 11), for bending angle, refractivity, dry pressure, dry geopotential height, dry temperature, and temperature. We show the results for five latitude zones and for the global mean at the full vertical grid, for bending angle up

to 60 km altitude, for refractivity up to 50 km, and for the other variables up to 40 km. At lower altitudes, we cut at 8 km for dry parameters, and at 2 km for temperature.

For CHAMP (Fig. 8), the structural uncertainty of trends from different processing centers is found small below 40 km for bending angle, below 30 km for refractivity and pressure, and below 25–28 km for (dry) temperature at all latitudes. Structural uncertainty increases above 25 km and at high latitudes, mainly due to increased sensitivity to the different bending angle

initialization approaches implemented at each center, including different high altitude background information. Compared to the results by Steiner et al. (2013) for CHAMP, we find in this study even better agreement between the centers because improved data versions have been delivered. At high latitudes the uncertainty is smaller here, which is most probably due to a new data version provided by GFZ.



For F3C (Fig. 9), the structural uncertainty is much smaller compared to CHAMP. It is low for bending angle up to 50 km, for refractivity up to 45 km, for pressure up to 40 km, and for (dry) temperature up to 30 km. Except at SHL, the structural uncertainty becomes larger above about 25 km altitude.

For GRACE (Fig. 10), the structural uncertainty is very small at all altitude levels and at all latitudes, except for SHL. Larger structural uncertainty is only found at upper altitudes for bending angle and refractivity and at SHL for all variables.

For Metop (Fig. 11), the structural uncertainty is found smallest compared to the other missions. High consistency is found at all latitudes and over all altitudes. A small difference in the trend near 20 km is visible for WEGC data. This is due to the handling of Metop data, where due to a tracking update in 2013 rising occultations are tracked only from about 20 km upwards.

A summary of the resulting standard deviation numbers is given in Fig. 12 for all parameters and all satellites. We set these results into context with the GCOS stability requirements for ECVs, defined by 0.05 K per decade air temperature in the troposphere and stratosphere (GCOS, 2016), formerly by 0.1 K/decade in the stratosphere (GCOS, 2011). For the other RO variables no dedicated GCOS requirements exist but they can be estimated from physical relations between these variables with reasonable scaling. The corresponding estimates for 0.1 K per decade in temperature are 0.05% per decade for refractivity (factor 0.5), 0.12% per decade for bending angle (factor ~2.4), 0.06% per decade for pressure, and about 4 m per decade for geopotential height. The relation between geopotential height and pressure changes is given via atmospheric scale height of about 70 m geopotential height change per one percent pressure change (see Steiner et al., 2013).

In Fig. 12, we visually relate the standard deviation to the GCOS stability criteria via color coding, where light orange indicates that the criteria are met for temperature with 0.05–0.1 K per decade and the corresponding criteria for the other RO variables. For the global average 90°S to 90°N, the standard deviation of bending angle trends is <0.06% for the altitude layers 8–25 km and 25–35 km, for all satellite missions. At high latitudes, it is larger for F3C at SHL (0.3%) and NHL (0.15%). For refractivity trends, the standard deviation is <0.03% at 8–25 km and at 25–35 km for all satellites globally. Only for CHAMP it is larger above 25 km and for Metop and GRACE at NHL/SHL (~0.05%). For (dry) pressure trends, the standard deviation is <0.03% at 8–25 km except at SHL for CHAMP. Dry geopotential height shows a standard deviation of <4 m for all satellites below 35 km, except at high latitudes. For CHAMP it is about 15–20 m at 25–35 km altitude.

For (dry) temperature trends, the standard deviation is <0.05 K at 8–25 km for all satellites, except for CHAMP it is <0.08 K. Even in the 25–35 km layer, the standard deviation for dry temperature is globally <0.05 K for Metop, and <0.1 K for F3C and GRACE whereas for CHAMP it is 0.5 K globally. Physical temperature shows even smaller uncertainty at lower altitudes, however, above about 30 km it can be larger than for dry temperature due to a priori information in moist air retrievals.

We find that RO multi-satellite data products from different centers are highly consistent at 8–25 km for all RO missions over all latitudes. Figure 12 reveals that F3C, GRACE and Metop are usable for climate studies up to higher altitudes, to 30–35 km for temperature, geopotential height and pressure, to 40 km for refractivity, and to 50 km for bending angle. There structural uncertainty meets the stringent stability requirements for air temperature as defined by GCOS (2016), as well as corresponding requirements for the other RO variables.





# 5 Summary and conclusions

The aim of this study was to assess the consistency and long-term stability of RO observations for use as climate data records of essential climate variables in a global climate observing system. We therefore performed a rigorous intercomparison study of a full set of RO data products from multiple satellites provided by different RO processing centers. We analyzed all available

RO data products from dry and moist air retrievals. The atmospheric variables included bending angle, optimized bending angle, refractivity, dry pressure, dry temperature, dry geopotential height, pressure, temperature, and specific humidity. Data products were delivered by five RO processing centers for the RO missions CHAMP, Formosat-3/COSMIC, GRACE, and Metop.

As a measure for consistency and stability, we investigated the structural uncertainty of RO multi-satellite records which arises

from different processing schemes. Based on the common subsets of delivered RO profiles, we computed de-seasonalized time series and difference time series of individual centers with respect to the all-center mean, as well as respective linear trends of the time series. The spread of the difference time series was investigated as one indication of structural uncertainty. We finally quantified the structural uncertainty of trends based on the standard deviation of the individual center trends. This uncertainty measure gives a representation of the stability of the multi-satellite RO records, enabling assessment against GCOS stability

requirements and of the consistency of products from different processing centers.

Globally, the standard deviation of bending angle trends is found <0.06% in the altitude layers 8–25 km and 25–35 km for all satellite missions. For refractivity trends, the standard deviation is <0.03% in these altitude layers for all satellites except CHAMP. For (dry) pressure trends, the standard deviation is <0.03% globally at 8–25 km, and <0.06% below 35 km depending on latitude. Dry geopotential height shows a standard deviation of <2–4 m below 35 km for all satellites except CHAMP. For

(dry) global temperature trends, the standard deviation is <0.05 K at 8–25 km and <0.1 K at 25–35 km for all satellites, except for CHAMP where it is <0.08 K and 0.5 K, respectively.

Our results show that RO multi-satellite data products from different centers are highly consistent between 8 and 25 km for all RO missions over all latitudes. Furthermore, data products from the newer satellite missions F3C, and specifically GRACE and Metop, are usable to higher altitudes due to advanced receivers (better onboard clocks) and lower bending angle noise at

higher altitudes. For these missions, (dry) temperature, dry geopotential height, and (dry) pressure is found to be consistent up to 30–35 km, refractivity up to 40 km, and bending angle up to 50 km.

In conclusion, we find that the RO record can be used for reliable climate trend assessments globally within 90°S to 90°N in these altitude regions, meeting the stringent GCOS stability requirements for air temperature and corresponding requirements for the other RO variables. Data users should be aware of the larger uncertainty of the CHAMP record at higher altitudes. Also,

temperature derived with a moist air retrieval can have a larger uncertainty above 25 km due to a priori information. Knowledge of the differences in quality of the various satellite data is essential, especially when data from several missions are combined into a multi-satellite record. The final Fig. 12 gives an instructive overview on the structural uncertainties for all RO variables over latitude and altitude.



This intercomparison study helped to further improve the maturity and quality of the RO records. During the course of work, we reported small issues and gave feedback to the processing centers, which was incorporated into the product development and resulted in improved reprocessed data sets for this study. We regard the quantification of structural uncertainty of multi-satellite RO records from different RO processing centers as an essential advance towards the establishment of a global climate

benchmark record as key component of GCOS.

Efforts at RO centers are ongoing on further improving and advancing RO data processing, such as the new WEGC RO processing system with integrated uncertainty propagation and traceability to the fundamental time standard (Li et al., 2015; Innerkofler et al., 2017; Schwarz et al., 2017; 2018; Gorbunov and Kirchengast, 2018). Also, new RO missions with advanced receivers will provide RO data with better quality. RO receivers are established on the Chinese FY-3 meteorological satellite

series (Sun et al., 2018), Metop-C is in orbit since November 2018, and the six-satellite FORMOSAT-7/COSMIC-2 constellation was successfully launched in June 2019 (Ho et al., 2019).

New receivers are capable of tracking different GNSS signals from the U.S. GPS, the Russian GLONASS, the European Galileo system, and the Chinese Beidou system and will provide a larger number of observations. These recently launched and further planned RO missions will ensure the continuation of the RO record into the future for long-term climate monitoring

and trend detection.

*Data availability*. Information on the availability of the analyzed data sets is given in Table 1.

*Author contributions*. A.K.S. designed the study, organized the data collection, discussed with the co-author team, analyzed

the results, and wrote the manuscript text. F.L. performed the computations and analysis, created all figures, and contributed to the paper text. C.O.A., H.G., S.-P.H., D.H, T.S. delivered the data products and provided information on data and processing description. A.J.M. influenced JPL's strategy on data products. All authors engaged in discussions on data characteristics and issues, on the interpretation of results and contributed to the finalization of the manuscript text.

*Competing interests*. The authors declare that they have no conflict of interest.

*Acknowledgements*. We thank B. Angerer (WEGC, AT), B. A. Iijima (JPL), and O. P. Verkhoglyadova (JPL) for valuable support on data processing and quality control of the provided data sets. We acknowledge M. Mochart (WEGC, AT) for his Master thesis work on a first analysis of the intercomparison study.

This work was funded by the Austrian Science Fund (FWF) under research grant P27724-NBL (VERTICLIM), the European Space Agency (ESA) project MMValRO, and the Austrian Research Promotion Agency (FFG-ALR) projects OPSCLIMVALUE (ASAP-11 848013) and ATROMSAF1 (ASAP-13 859771). Portions of this research were carried out at the Jet Propulsion Laboratory, California Institute of Technology, under a contract with the National Aeronautics and Space Administration. Support of the NASA Earth Science Division is gratefully acknowledged. H. Gleisner (DMI), J. K. Nielsen



(DMI), K. B. Lauritsen (DMI), and S. Syndergaard (DMI) were supported by the Radio Occultation Meteorology Satellite Application Facility (ROM SAF) which is a decentralized operational RO processing center under EUMETSAT. The radio occultation data from DMI were provided by the ROM SAF (http://www.romsaf.org). The manuscript contents are solely the opinions of the author(s) and do not constitute a statement of policy, decision, or position on behalf of NOAA or the U.S.

Government.

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



**Tables:**

**Table 1.** Overview on processing steps for RO dry and moist air retrieval at DMI, GFZ, JPL, UCAR, WEGC.

| Processing step | Center | Implementations of each center |
|---|---|---|
| **URL** | DMI | http://www.romsaf.org |
| | GFZ | http://www.gfz-potsdam.de/en/section/space-geodetic-techniques/topics/gnss-radio-occultation/ |
| | JPL | https://genesis.jpl.nasa.gov/genesis/ |
| | UCAR | http://cdaac-www.cosmic.ucar.edu |
| | WEGC | http://www.wegcenter.at |
| **Processing version and POD phase and orbit data version** | DMI | GPAC-2.3.0/ROPP software; Excess phase, amplitude and orbit data from UCAR. |
| | GFZ | Version POCS ATM vers.006; POD: EPOS-OC, RSO orbit products (König et al., 2006); Excess phase: CHAMP: Single differencing, reference link smoothing; GRACE: zero differencing. |
| | JPL | Version 2.7 (single differencing, cubic phase smoothing); POD: GPS orbits from JPL FLINN products; LEO orbits reduced-dynamic strategy using GIPSY software (Bertiger et al., 1994). |
| | UCAR | CDAAC version 4.6; Bernese version 5.2. |
| | WEGC | OPSv5.6; UCAR/CDAAC orbit and phase data (Angerer et al., 2017, Table 1). |
| **Calculation of bending angle (BA)** | DMI | Canonical Transform (CT2) inversion <20 km (Gorbunov and Lauritsen, 2004), transition to geometric optics (GO) inversion at 20–25 km, GO >25 km. |
| | GFZ | Full Spectrum Inversion (FSI) <15 km (Jensen et al., 2003), smooth transition between 11 km and 15 km to GO, GO >15 km. |
| | JPL | Canonical transform (CT) after (Gorbunov, 2002) applied to L1 at impact height <30 km. GO for L1 >30 km and L2 at all heights. |
| | UCAR | Phase matching <20 km (Jensen et al., 2004), GO >20 km. |
| | WEGC | CT2 inversion (Gorbunov and Lauritsen, 2004) with a Gaussian transition of 4.5 km width and variable center height between 7 km and 13 km, GO above. |
| **Ionospheric correction** | All | Linear combination of L1 and L2 BA (Vorob'ev and Krasil'nikova, 1994). |
| | DMI | Linear combination, ionospheric correction extrapolated with constant L1–L2 BA below dynamic L2 height – transition over 2 km. |
| | GFZ | Linear combination, ionospheric correction extrapolated with constant L1–L2 BA below 12 km. |
| | JPL | Linear combination, ionospheric corr. term extrapolation <10 km when L2 1sec SNR<30 V/V. |
| | UCAR | Above 20 km: correction of L1 BA by L1–L2 BA smoothed with window determined individually for each occultation to minimize combined noise (Sokolovskiy et al., 2009). Below 20 km: L1 BA corrected by a 3-parameter function fitted to observational L1–L2 BA at 20–80 km (Zeng et al., 2016). |
| | WEGC | Linear combination, ionospheric correction term extrapolated with constant L1–L2 BA <15 km. |





| **Initialization of BA** | DMI | Optimization with dynamic estimation of observation errors (Gorbunov, 2002) and background errors fixed at 50%, background based on BAROCLIM (best global fit to data between 40 and 60 km, scaled using two-parameter regression) (Scherllin-Pirscher et al., 2015). |
| | GFZ | Optimization after (Sokolovskiy and Hunt, 1996) with MSISE-90 (>40 km), observation error variance estimated as 25% of mean observation-background deviation between 60 km and 70 km. |
| | JPL | Exponential function fit at 50–60 km and extrapolation >60 km impact height. |
| | UCAR | Static optimization (independent of the observational noise), 2-parameter fitting of NCAR BA climatology (Randel et al., 2002) to observational BA in 35–60 km interval, transition to fitted BA climatology in the same interval, transition to un-fitted BA climatology in the 55–65 km interval. |
| | WEGC | Optimization >30 km with ECMWF short-range forecasts (24h or 30h) and above with MSISE-90 to 120 km, dynamic estimation of observation errors and inverse covariance weighting (Schwärz et al., 2016, Appendix A.4). |
| **Refractivity Retrieval** | All | Abel inversion (Fjeldbo et al., 1971) of optimized bending angle profile. |
| | DMI | Abel inversion below 150 km. |
| | GFZ | Abel Inversion below 150 km. |
| | JPL | Abel Inversion below 120 km. |
| | UCAR | Abel inversion below 150 km. |
| | WEGC | Abel inversion below 120 km. |
| **Dry air retrieval** | All | Refractivity (N) is directly proportional to air density (ideal gas equation). |
| | DMI | Pressure integration, hydrostatic integral initialization at 150 km, upper boundary condition from refractivity gradient, geopotential height relative to EGM-96 geoid. |
| | GFZ | Hydrostatic integral initialization at 100 km with MSISE-90 pressure, geopotential height relative to EGM-96. |
| | JPL | Hydrostatic integral initialization at 40 km using ECMWF analysis, geopotential height relative to JGM-3. |
| | UCAR | Hydrostatic integral initialization at 150 km with zero boundary condition. |
| | WEGC | Hydrostatic integral initialization at 120 km with MSISE-90 pressure, geopotential height relative to EGM-96. |
| | All | Dry temperature ($T_d$) is obtained using the Smith-Weintraub formula for dry air (Smith and Weintraub, 1953) and the equation of state (ideal gas). |
| **Moist air retrieval** | DMI | 1D-Var using ERA-Interim as background and refractivity observations as input. |
| | GFZ | Not included, but relevant data products can be provided on demand |
| | JPL | Direct method using temperature from ECMWF when T >240 K (Kursinski et al., 1996). |
| | UCAR | 1D-Var using ERA-Interim as background and refractivity observations (Wee, 2005). |
| | WEGC | Above 16 km: calculation of physical temperature $T$ and pressure $p$ using a first order approximation for the ratio between $p$ and dry pressure $p_d$. Below 14 km, with sinusoidal transition between 16–14 km: -retrieval of T and p using ECMWF SR-FC specific humidity $q_B$ -retrieval of q and p using ECMWF SR-FC temperature $T_B$ -statistical optimization of $T$ and $q$ with $q_B$ and $T_B$, background error from ROPPv6.0 (Culverwell and Healy, 2011), RO observational error (Scherllin-Pirscher et al., 2011a) |





| | | |
|---|---|---|
| **Quality control (QC)** | DMI | Provider QC (reject if phase data are flagged);<br>QC of L2 quality from impact parameters (reject if noise is too large);<br>QC of BA using ERA-Interim forecasts (reject if >90% in 10–40 km);<br>QC of regression parameters (reject if too far from 1.0);<br>QC of optimized BA using background (reject if >5 µrad above 60 km);<br>QC of background weight in optimization (reject if >10% below 40km);<br>QC of refractivity using ERA-Interim forecasts (reject if >10% in 10–35 km);<br>QC of dry temperature using ERA-Interim forecasts (reject if >20 K in 30–40 km);<br>QC of 1D-Var cost function (reject if too large);<br>QC of 1D-Var convergence (reject if too many iterations). |
| | GFZ | Minimum duration of occultation event: 20 s; Quotient L1/L2 excess phase forward differences between 0.97 and 1.03 for at least 650 connected data samples<br>QC of refractivity $N$ using MSISE-90: reject if $\Delta N$ >22.5% between 8–31 km. |
| | JPL | Refractivity difference with ECMWF <10% between 0–40 km and temperature difference with ECMWF <10 K below 40 km. |
| | UCAR | Multiple QC checks including:<br>- Comparison of retrieved $N$ and $N$ from NCAR climatology (Randel et al. 2002);<br>- Comparison of maximum relative BA difference between RO and NCAR climatology;<br>- BA error check of local spectral width; - SNR too low;<br>- Check of L2 data quality by comparison of maximum L1–L2 Doppler;<br>- Checks of mean and standard deviation of difference of retrieved and climatological BA between 60–80 km. |
| | WEGC | Raw QC check: Straight line tangent point altitude (SLTA) range at least between 65–20 km;<br>GO only QC of BA: - cut off <15 km impact height if gradient is too large;<br>- reject if BA <0 rad below 50 km; - reject if bias relative to MSIS-90 >$10^{-5}$ rad;<br>- reject if standard deviation relative to MSIS-90 >$5*10^{-5}$ rad;<br>WO only QC: cut off data at bottom of measurement if:<br>- amplitude of CA-signal is lower than 10% of max amplitude;<br>- smoothed GO BA (over 3 km) exceeds 0.05 rad;<br>- smoothed impact parameter (over 3 km) <0 m;- SLTA < –250 km.<br>QC of BA, $N$, $T$ using ECMWF analyses: reject if $\Delta$BA >20%, $\Delta N$ > 10% in 5–35 km or $\Delta T$ >20 K in 8–25 km |
| **Reference frame vertical coordinate** | DMI | Earth figure: WGS-84 ellipsoid; Vertical coordinate: mean-sea level (MSL) altitude; Conversion of ellipsoidal height to MSL altitude (at SLTA=0 TP location) via EGM-96 geoid smoothed to 1° x 1° resolution. |
| | GFZ | Earth figure: WGS-84 ellipsoid, EGM-96 geoid used for altitude above MSL calculation. |
| | JPL | Earth figure: IERS Standards 1989 ellipsoid; Vertical coordinate: MSL altitude computed using the JGM3/OSU91A geoid truncation at spherical harmonic degree 36. |
| | UCAR | Earth figure: WGS-84 ellipsoid; The occultation point is determined using BA for CIRA+Q climatology (Kirchengast et al., 1999) and 500 m observed excess phase. The center of reference frame is in the local center of curvature of the reference ellipsoid at the occultation point (Syndergaard, 1998) in the direction of the occultation plane. Impact height (for BA) and height (for $N$, $p$, $T$) are obtained by subtracting local curvature radius and JGM2 geoid undulation (at the occultation point) from the impact parameter and radius. |
| | WEGC | Earth figure: WGS-84 ellipsoid; Vertical coordinate: MSL altitude; Conversion of ellipsoidal height to MSL altitude (at SLTA=0 TP location) via EGM96 at 0.25° x 0.25° resolution. |





| Reference/ Publication | DMI | ROM SAF ATBD documents: http://www.romsaf.org/product_archive.php |
| | GFZ | ftp://isdcftp.gfz-potsdam.de/ |
| | JPL | Hajj et al. (2002) |
| | UCAR | CDAAC Website documentation area: http://cdaac-www.cosmic.ucar.edu/cdaac/doc/overview.html |
| | WEGC | https://doi.org/10.25364/WEGC/OPS5.6:2019.1; Schwärz et al. (2016), Angerer et al. (2017) |



**Table 2.** Overview on RO data used in this study: processing center, satellite mission, period, variables.

| Center | Satellites | Period | Variables |
|---|---|---|---|
| DMI | CHAMP | 09/2001–09/2008 | All variables[1] |
| | COSMIC | 05/2006–12/2016 | All variables |
| | GRACE | 03/2007–12/2016 | All variables |
| | METOP | 02/2008–12/2016 | All variables |
| GFZ | CHAMP | 05/2001–09/2008 | All except: $p$, $T$, $q$ |
| | GRACE | 02/2006–11/2017 | All except: $p$, $T$, $q$ |
| JPL | CHAMP | 04/2001–09/2008 | All variables |
| | COSMIC | 05/2006–12/2016 | All variables |
| UCAR | CHAMP | 05/2001–09/2008 | All except: $\alpha_{opt}$ |
| | COSMIC | 05/2006–04/2014 | All except: $\alpha_{opt}$ |
| | METOP | 02/2008–12/2015 | All except: $\alpha_{opt}$ |
| WEGC | CHAMP | 05/2001–09/2008 | All variables |
| | COSMIC | 08/2006–12/2018 | All variables |
| | GRACE | 03/2007–11/2017 | All variables |
| | METOP | 02/2008–12/2018 | All variables |
| All centers common periods | CHAMP | 09/2001–09/2008 | |
| | COSMIC | 08/2006–04/2014 | |
| | GRACE | 03/2007–12/2016 | |
| | METOP | 03/2008–12/2015 | |

[1] All variables include $\alpha$, $\alpha_{opt}$, $N$, $p_{dry}$, $T_{dry}$, $Z_{dry}$, $p$, $T$, $q$.





**Figures:**

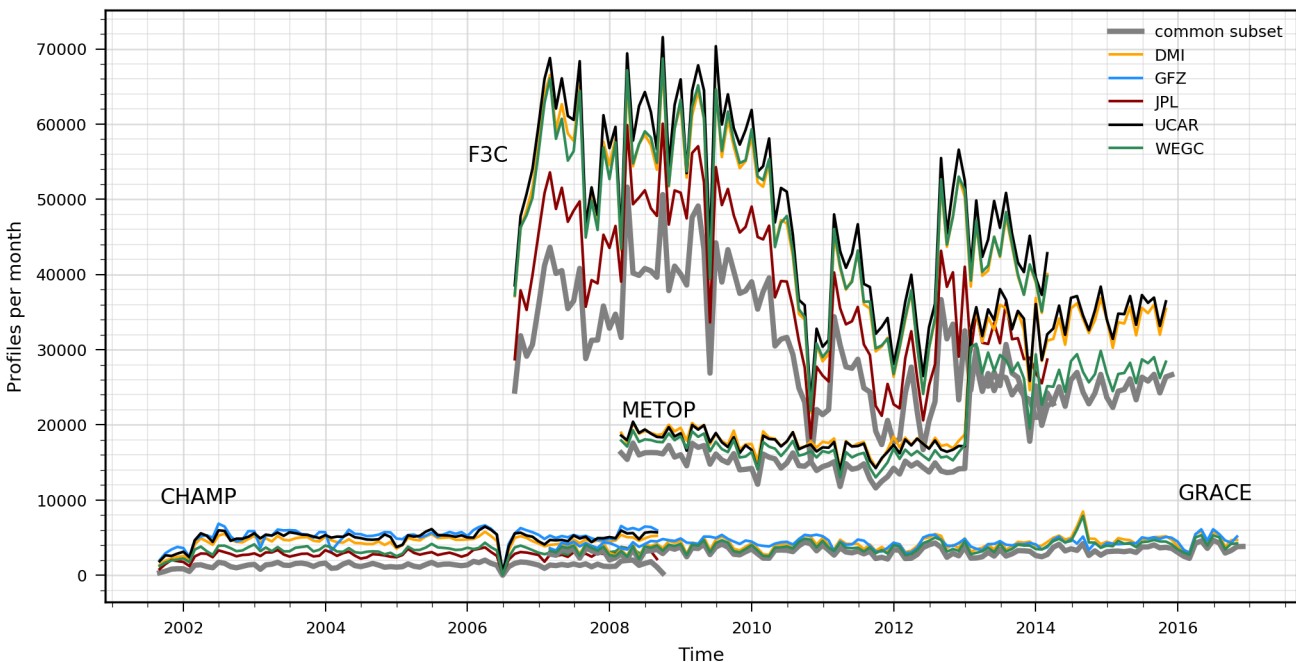

**Figure 1.** Number of RO profiles per month delivered by each processing center, DMI (yellow), GFZ (blue), JPL (red), UCAR

5 (black), WEGC (green), and the maximum subset of profiles (gray), shown for the respective time periods of the four missions

CHAMP, F3/C, GRACE, and Metop.





**Figure 2.** Global mean difference of atmospheric profiles from each center to the all-center mean for on exemplary month (July 2008) shown for the satellite missions, CHAMP, COSMIC, GRACE, and Metop (left to right) for bending angle, optimized bending angle, refractivity, dry temperature, and temperature (top to bottom). The number of data points is shown in the left subpanels.





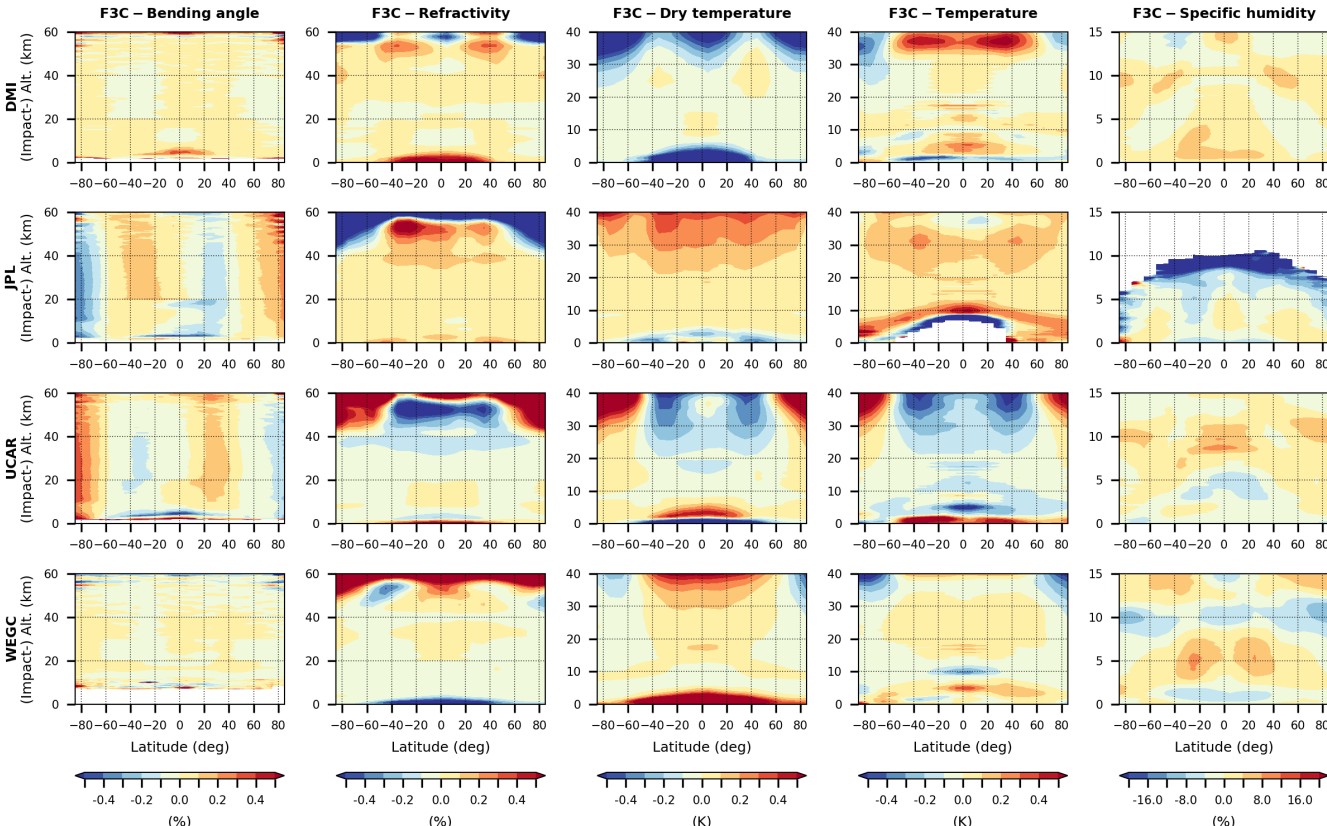

**Figure 3.** Mean difference of each center, DMI, JPL, UCAR, WEGC (top to bottom), to the all-center mean for F3C data averaged over 08/2006–04/2014, shown for bending angle, refractivity, dry temperature, temperature, and specific humidity (left to right).





(a)

(b)

**Figure 4.** CHAMP bending angle **(a)** and dry temperature **(b)**: De-seasonalized anomaly difference time series of each center to the all-center mean for latitude bands 90°S to 60°S, 20°S to 20°N, 60°S to 90°S and globally 90°S to 90°N (left to right) for altitude layers 8–18 km, 18–25 km, 25–30 km (bottom to top). Time series from DMI (orange), GFZ (blue), JPL (red), UCAR (black), and WEGC (green) are shown.







**Figure 5.** F3C bending angle **(a)** and dry temperature **(b)**: De-seasonalized anomaly difference time series of each center to the all-center mean for latitude bands 90°S to 60°S, 20°S to 20°N, 60°S to 90°S and globally 90°S to 90°N (left to right) for altitude layers 8–18 km, 18–25 km, 25–30 km (bottom to top). Time series from DMI (orange), JPL (red), UCAR (black), and WEGC (green) are shown.





(a)

(b)

**Figure 6.** GRACE bending angle **(a)** and dry temperature **(b)**: De-seasonalized anomaly difference time series of each center to the all-center mean for latitude bands 90°S to 60°S, 20°S to 20°N, 60°S to 90°S and globally 90°S to 90°N (left to right) for altitude layers 8–18 km, 18–25 km, 25–30 km (bottom to top). Time series from DMI (orange), GFZ (blue), and WEGC (green) are shown.





(a)

(b)

**Figure 7.** Metop bending angle **(a)** and dry temperature **(b)**: De-seasonalized anomaly difference time series of each center to the all-center mean for latitude bands 90°S to 60°S, 20°S to 20°N, 60°S to 90°S and globally 90°S to 90°N (left to right) for altitude layers 8–18 km, 18–25 km, 25–30 km (bottom to top). Time series from DMI (orange), UCAR (black), and WEGC (green) are shown.



**Figure 8.** CHAMP structural uncertainty indicated as the standard deviation (gray) of the individual center trends per decade, for DMI (orange), GFZ (blue), JPL (red), UCAR (black), and WEGC (green) shown for bending angle, refractivity, dry pressure, geopotential height, dry temperature, and temperature (top to bottom). The all-center mean trend profile (bold black line) and the altitude-layer mean trends (crosses) are indicated.

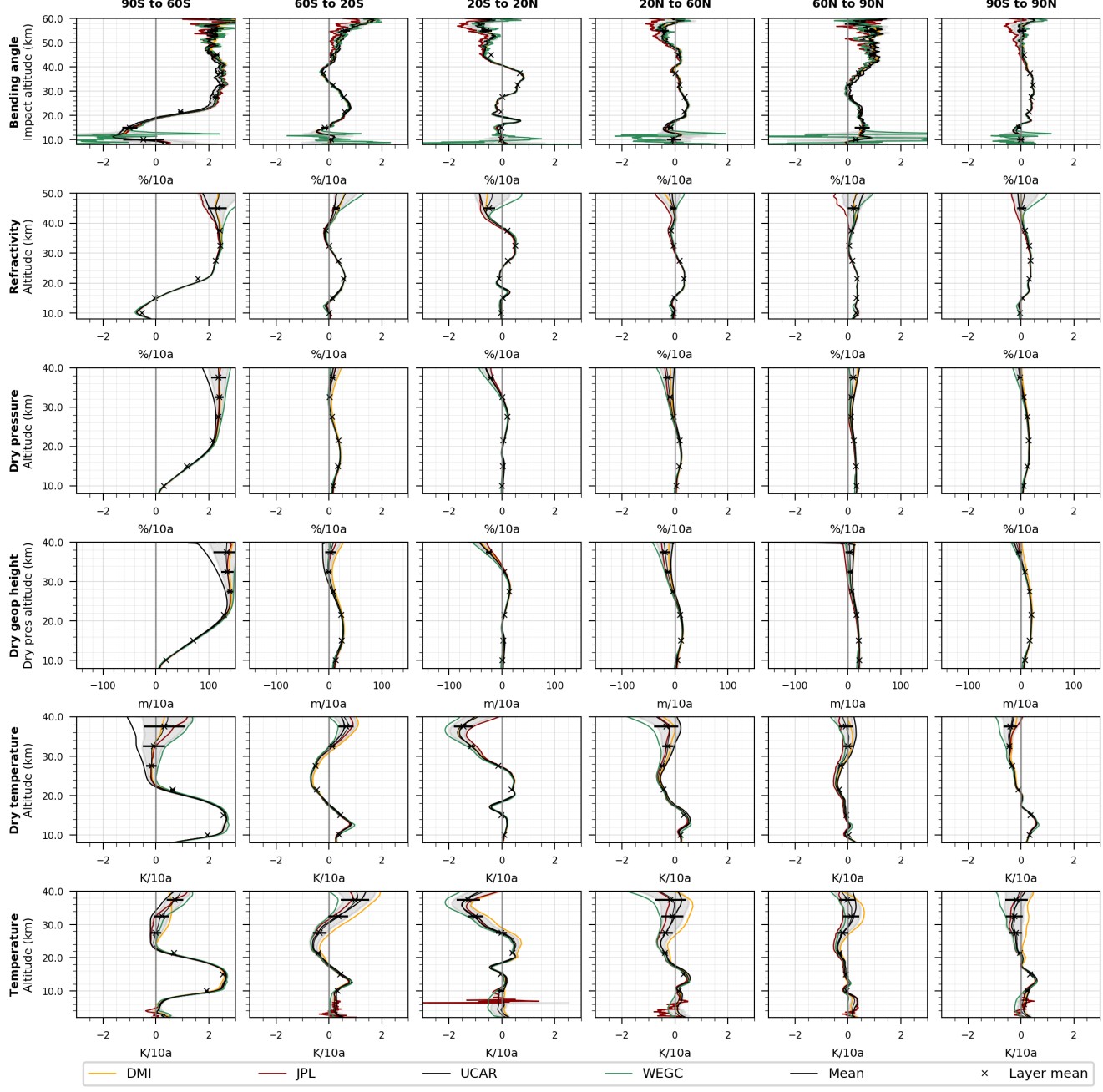

**Figure 9.** F3C structural uncertainty indicated as the standard deviation (gray) of the individual center trends per decade, for DMI (orange), JPL (red), UCAR (black), and WEGC (green) shown for bending angle, refractivity, dry pressure, geopotential height, dry temperature, and temperature (top to bottom). The all-center mean trend profile (bold black line) and the altitude-layer mean trends (crosses) are indicated. Note that for bending angle the median rather than mean trend of all centers is indicated, due to some outlier profiles in the JPL data.



**Figure 10.** GRACE structural uncertainty indicated as the standard deviation (gray) of the individual center trends per decade, for DMI (orange), GFZ (blue), and WEGC (green) shown for bending angle, refractivity, dry pressure, geopotential height, dry temperature, and temperature (top to bottom). The all-center mean trend profile (bold black line) and the altitude-layer mean trends (crosses) are indicated.







**Figure 11.** Metop structural uncertainty indicated as the standard deviation (gray) of the individual center trends per decade, for DMI (orange), UCAR (black), and WEGC (green) shown for bending angle, refractivity, dry pressure, geopotential height, dry temperature, and temperature (top to bottom). The all-center mean trend profile (bold black line) and the altitude-layer mean trends (crosses) are indicated.





**Figure 12.** Overview on structural uncertainty for different RO missions, CHAMP, F3C, GRACE, and Metop (left to right). Shown is the standard deviation of individual center trends per decade, for RO bending angle, refractivity, dry pressure, geopotential height, dry temperature, and temperature (top to bottom), for all latitude zones and altitude layers in the sub-panels.