# Peer review of "Consistency and structural uncertainty of multi-mission GPS radio occultation records"

_Atmospheric Measurement Techniques, 2019_

## Referee Comment (RC1) · Anonymous Referee #2 · 8 Dec 2019

I thank the authors for a manuscript that has obviously been polished by thorough proof-reading before submission. This paper represents the tip of the iceberg of what seems to be a large community undertaking. It is quite nice to see such synchronization and coordination to advance the science. I have one major comment on the paper. Other detailed comments are indicated thereafter.

**Major comment**

It is unclear what structural uncertainty encompasses, and how a standard deviation computed from 5 different products can say everything about the confidence one should place in these products for climate monitoring. Furthermore, it is unclear how using this metric is applicable, alone and by itself, to assert compliance to the GCOS requirements for stability. For example, if all data producers used exactly the same processing techniques, one would expect to see a collapse in the product spread between the various products; would this give us absolute confidence in the real stability of the instruments? (one is allowed to doubt)

**Detailed comments**

Did all current RO data processing centers take part in this exercise, especially those centers processing long time-series and third-party missions? If not, it may be useful to indicate if future work will strive to include those other centers.

"Structural uncertainty" is neither defined nor referenced in this paper. It is a central concept to this paper, and one not defined in textbooks or standards such as the BIPM Guide to the Expression of Uncertainty in Measurement (GUM). It cannot be expected for readers to guess what this particular concept of uncertainty means, or what it represents.

Section 2 is highly informative and packs a lot of information. My feeling is that the presentation of the data (as provided by the various centers) and the presentation of the methodology (for comparing these data) need to be separated. It would be fitting to split the section accordingly (e.g., Section 2: Data, Section 3: Analysis methodology, or equivalent). This would avoid a potential confusion between radio occultation data processing (done outside of this study), and the analysis of the results (as conducted and presented in the paper).

The section on data could benefit from being reorganized as follows: - Starting from the raw data, - Proceeding to the higher-level products /retrievals (without going back at the end to discuss clock errors etc.), - Presenting, at each step, the commonalities for all centers, before indicating the differences (e.g., center X did not produce ...).

"as three centers start with the same phase and orbit data, RO products are not independent": I do not understand why this statement only applies to these 3 centers, and not all of them. (The various RO processing centers, for each given mission, do start from the same receiver data?)

The subscript s refers to the satellite receiver (not transmitter). It may be useful to indicate 'receiver'.

Were equations (1) to (6) applied to the subset of common profiles processed by all centers?

"only JPL provided a smaller amount": Looking at figure 1, one sees that JPL did provide a smaller amount indeed, of about 10,000 profiles per month, compared to the pack of other producers. However, quite interestingly, the common subset of profiles, between all producers, is also 10,000 lower than the JPL count. This near-match in the differences (from other producers to JPL, and then from JPL to common subset) is quite puzzling. Could it come from an unexpected issue in individual ID assignment (e.g., a shift by a minute or so), which would make many JPL profiles not match the other IDs?

"we are interested in the structural uncertainty of trends represented by the standard deviation of the n\_center individual center trends": Unless I misunderstood something, given that n\_centers is (at most) 5, this means that the central metric of the paper is a standard deviation based on a population of 5 members. How reliable is a standard deviation based on so few members? This needs to be discussed.

In the future, wouldn't there be a more robust estimate that can computed, to characterize this spread, or inconsistencies, given a such small sample?

The fact that the spread in physical temperatures is reduced so much, from the spread in dry temperatures, needs to be explained. Does it point to the fact that the products use similar (background) constraints in the retrieval, and then correspondingly that all products are probably quite representative of these constraints?

The difference found at high latitudes is one very interesting result of this paper. This is mentioned as being related to Arctic SSW (something which correlates well by looking

СЗ

at the 60N-90N timeseries and the occurrences of peaks in winter). I would think this deserves a separate sub-section in discussion, with additional results to go a bit further. Is it possible to illustrate the influence of the different strategies for high-altitude initialization in these situations, e.g., by picking a particular SSW event, and showing individual profiles?

Throughout the paper, all statements making the link to the GCOS stability requirement need to be revised, as they all fail to include the other sources of uncertainty affecting stability (other than differences in processing).

Typo: 'on exemplary' -> one exemplary

Table 1, impact height is only defined for UCAR. Do the other centers use different definitions? Shouldn't this have been the same definition for all? In other terms, isn't there a RO community-approved definition of 'impact height'?

Table 1 indicates several vertical reference frames, not always WGS-84 ellipsoid and EGM-96 geoid. As a reminder, WMO Executive Council 59 (in 2007) adopted a draft resolution proposed by the Commission for Basic Systems of these two elements (WGS-84 and EGM-96) as the fundamental bases for vertically referencing all station observations. This choice was also relayed in the coordinated satellite community for geostationary products, by CGMS in 2011 ("LRIT/HRIT Global Specification"). It is not so much a matter of choosing the 'best' reference frame for each observing system, but one that is fit for purpose and a unique standard in a community, so as to avoid introducing artificial discrepancies/differences. It could hence be useful to make a note that some of the differences between data producers presented in Table 1, such as this one, will eventually need resolving.

Figure 1, it is unclear why the number of points differ by altitude, even though the list of profiles is supposedly common for all centers. Is this caused by different QC at each vertical level?

Figures 4 to 7, the equations (shown in the legend inside each plot) are too small to be legible; they could go into a new table, or, better yet, be summarized in a graphic, in a similar form as Fig. 12.

Figure 8 to 11 pack, in total, over 700 vertical profiles. Surely, there must be a way to summarize this into a manageable amount of information, for readers to grasp the message. These plots are surely of value in a supplement, though.

---

## Referee Comment (RC2) · Sean Healy (Referee) · 3 Jan 2020

Consistency and structural uncertainty of multi-mission GPS radio occultation records.

by Steiner et al

**General Comments**

This paper investigates the consistency of climate data records retrieved from GPS radio occulation measurements, produced by a number of processing centres. It is a very thorough piece of work, and it is generally well explained. Overall, it will be a useful addition to the literature. I recommend publish subject to the minor revisions detailed below.

Specific comments

Section 1

Page 2, Line 2: Suggest inserting "in situ" before observations, i.e.  $\Rightarrow$  " in situ observations" because it could be argued that there are many satellite radiances available.

Page 2, Line 15: the accuracy requirements, 0.1 K for climate and 1 K for NWP, need further explanation. The 1 K for NWP is presumably a random error for a given observation, but what is the definition of the 0.1 K requirement? Further, on Page 3, line 25 it says the observational uncertainty estimate for an individual RO observation is 0.7 K near the tropopause. Is this consistent with the 0.1 K climate requirement? Please clarify. Similarly, clarify "measurement uncertainty of 0.5 K" on Page 2, line 19.

**Section 2**

Page 5, Line 18: "Two coherent carrier signals ...". This sentence may give the impression that the ionospheric correction is in phase space. Please clarify.

Page 6, Line 2: It probably should be noted that no centre is currently trying to correct residual ionospheric errors using, for example, techniques such as those in Danzer et al (2015). Although there is still work required to demonstrate this approach (Danzer et al, 2019 submitted), it should be noted that redidual ionospheric errors are a potentially a common error at all the centres.

Page 6, Line 9: Some NWP centres have moved away from Smith and Weintraub (1953) to potentially more accurate formulations including both updates to the assumed C02 concentration and non-ideal gas effects. This is mainly as a result of work by Dr Aparicio. See Appicio and Larosche (2011) and references therein, Cucurull et al (2013), Healy (2011). The NWP implementations should be noted.

Figure 2: The Metop bending angles for WEGC at  ${\sim}15$  km seem to be an outlier. Any reason for this?

Figure 3: The JPL and UCAR appear to have almost equal and opposite bending angle biases. Please discuss.

Page 10, line 21: "Above this altitude, WEGC ...". It might be worth adding that the WEGC dry-temp and temperature differences above 16 km shown in Figure 3 are because of different all centre mean values.

Section 4.2, Page 11, Lines 9-10. "Larger variability ..." for JPL is likely due to bending angle extrapolation? Why is extrapolation relevant here?

Section 5

Page 14, line 16. When quoting the uncertainty in the trends , e.g. "0.06 %", include "per decade".

Page 14, line 26. The bending angles are found to be consistent up to 50 km because they are less sensitive to a priori information. Ringer and Healy (2006) suggested monitoring the climate in bending angle space for this reason, althought the interpretation of bending angle trends is more complicated. Consider adding this reference.

Technical suggestions

The text on many figures is still very difficult to read.

Figure 4b, 5b, 6b, 7b, 8-18km dry temperature time series. The vertical ranges/axes could be expanded.

Figure 6a, 7a. Better vertical ranges could be used in these figures.

**Suggested References**

Cucurull, L., Derber, J. C., and Purser, R. J. (2013), A bending angle forward operator for global positioning system radio occultation measurements, J. Geophys. Res. Atmos., 118, 14-28, doi:10.1029/2012JD017782.

Aparicio, J. M., and Laroche, S. (2011), An evaluation of the expression of

the atmospheric refractivity for GPS signals, J. Geophys. Res., 116, D11104, doi:10.1029/2010JD015214.

Healy, S. B. (2011), Refractivity coefficients used in the assimilation of GPS radio occultation measurements, J. Geophys. Res., 116, D01106, doi:10.1029/2010JD014013.

Ringer, M. A., and Healy, S. B. (2008), Monitoring twenty-first century climate using GPS radio occultation bending angles, Geophys. Res. Lett., 35, L05708, doi:10.1029/2007GL032462.

Danzer, J., Healy, S. B., and Culverwell, I. D.: A simulation study with a new residual ionospheric error model for GPS radio occultation climatologies, Atmos. Meas. Tech., 8, 3395\u20133404, https://doi.org/10.5194/amt-8-3395-2015 2015.

---

## Author Comment (AC1) · 24 Feb 2020

We thank the reviewer for the constructive comments and questions. We will revise the manuscript accordingly. Please find our responses to all comments below.

**Major comment

It is unclear what structural uncertainty encompasses, and how a standard deviation computed from different products can say everything about the confidence one should place in these products for climate monitoring.  Furthermore, it is unclear how using this metric is applicable, alone and by itself, to assert compliance to the GCOS requirements for stability.  For example, if all data producers used exactly the same processing techniques, one would expect to see a collapse in the product spread be-

tween the various products; would this give us absolute confidence in the real stability of the instruments? (one is allowed to doubt)

*Response

GPS RO is regarded a stable technique providing benchmark observations (Goody et al. 1998; GCOS 2011). Traceability to standards of the international system of units (SI) is the foundation for benchmark measurements (e.g., Leroy et al., 2006; Ohring, 2007), i.e., they are related to absolute standards and may be repeated at any subsequent time with precise comparability (Goody et al., 1998; 2002). As GPS RO measurements are based on highly precise and stable atomic clocks, long-term frequency stability is ensured. The fundamental measurement is the GNSS signal phase change as function of time, measured with high precision, solely in terms of timing, and is an absolute measurement and SI-traceable (e.g., Ohring, 2007). RO bending angle is given as one example of a fundamental climate data record by GCOS (2016). In this respect, we investigate the consistency of RO time series from different data centers. We assess if final data products from different processing systems (based on the same raw measurements) are consistent and if they give consistent trend information. Therefore, we estimate the structural uncertainty, which arises due to different choices in processing and methodological approaches for constructing a data set from the same raw data (Thorne, 2005). As a measure for structural uncertainty, we analyze the spread in difference time series and the spread in trends (Wigley, 2006). Analyzing difference time series removes the variability that is common to the data sets and isolates those differences that may be due to differences in data set production methods, i.e., structural uncertainty of the data records. Correspondingly, we chose the standard deviation of trends as a measure of the structural uncertainty in trends. The challenge is to quantify the true spread given a limited number of data sets. Thus, it is important to have several independent data records, and at least a minimum of three independently processed data sets (Thorne, 2005).

As these points are also addressed in the detailed comments below, please find our

further detailed responses there, at Comment 2 on structural uncertainty, Comment 8 on standard deviation, and Comment 11 on stability.

\*\*Detailed comments

\*Comment 1

Did all current RO data processing centers take part in this exercise, especially those centers processing long time-series and third-party missions? If not, it may be useful to indicate if future work will strive to include those other centers.

\*Response 1

Yes, all current RO data processing centers that are processing several or all available RO missions and that provide RO data for long-term records (from CHAMP to current RO missions) took part in this study. As explained in the introduction section of the manuscript, this is an RO community effort ongoing since 2006 (http://irowg.org/projects/rotrends/). All international RO processing centers collaborate on the intercomparison of RO data with the common objective of a continuous improvement of the maturity of RO data products for use in atmosphere and climate studies. We specify this in the introduction section of the revised manuscript: "We systematically intercompare RO data products provided by those five international RO processing centers that are processing several or all available RO missions and that provide RO data for long-term records (from CHAMP to current RO missions)."

\*Comment 2

"Structural uncertainty" is neither defined nor referenced in this paper. It is a central concept to this paper, and one not defined in textbooks or standards such as the BIPM Guide to the Expression of Uncertainty in Measurement (GUM). It cannot be expected for readers to guess what this particular concept of uncertainty means, or what it represents.

\*Response 2

As in Steiner et al. (2013), we use the concept of structural uncertainty discussed by Thorne (2005) and Wigley (2006) with respect to uncertainties in climate trends. Structural uncertainty of a specific observational record arises due to different choices in processing and methodological approaches for constructing a data set from the same raw data (Thorne, 2005). The challenge is thus to quantify the true spread of physically possible solutions, given a limited number of data sets. The more independent data records the better, but at least three independently derived datasets are regarded necessary for an estimate of the structural uncertainty (Thorne, 2005). We included a paragraph in the introduction section of the revised manuscript on explaining structural uncertainty: "Structural uncertainty of an observational record arises due to different choices in processing and methodological approaches for constructing a data set from the same raw data (Thorne, 2005). The challenge is thus to quantify the true spread of physically possible solutions from a limited number of data sets. At least three independently derived datasets are regarded necessary for an estimate of the structural uncertainty, but the more data sets the better. Thus, multiple independent efforts should be undertaken to create climate records."

*Comment 3

Section 2 is highly informative and packs a lot of information. My feeling is that the presentation of the data (as provided by the various centers) and the presentation of the methodology (for comparing these data) need to be separated. It would be fitting to split the section accordingly (e.g., Section 2: Data, Section 3: Analysis methodology, or equivalent). This would avoid a potential confusion between radio occultation data processing (done outside of this study), and the analysis of the results (as conducted and presented in the paper). The section on data could benefit from being reorganized as follows: - Starting from the raw data, - Proceeding to the higher-level products /retrievals (without going back at the end to discuss clock errors etc.), - Presenting, at each step, the commonalities for all centers, before indicating the differences (e.g., center X did not produce . . .).

*Response 3

Thanks to the reviewer for these suggestions, which is actually close to the already implemented structure. Section 2 gives a description of RO data and of the data processing at the centers (done outside this study). Section 3 gives a description of the study set up and method used in this study. In Section 4, we present and discuss the results. In the revised manuscript, we further improved the structure. We structured section 2 into three subsections as follows. 2.1 Radio occultation missions and data In this subsection, we describe the occultation missions and data used in this study. 2.2 General RO data processing description In this subsection, we describe the basic retrieval steps, which are common for each center, starting from phase measurements to atmospheric bending angle and to atmospheric variables for dry and moist atmospheric conditions. 2.3 Center-specific RO processing steps and comparison In this subsection, we explicitly discuss the center-specific processing steps and the main differences between the centers. A complete and concise overview is given in Table 1, which shows commonalities and differences in the processing steps of each center. For section 3, we adjusted the title to make it clearer. It now reads: 3 Study setup and analysis method

*Comment 4

"as three centers start with the same phase and orbit data, RO products are not independent": I do not understand why this statement only applies to these 3 centers, and not all of them. (The various RO processing centers, for each given mission, do start from the same receiver data?)

*Response 4

All centers use the same receiver data sets ( i.e. data from missions CHAMP, F3C, GRACE, METOP). The structural uncertainty involved in the raw data (e.g, firmware coding) is hard to address. In this study, we focus on analyzing the structural uncertainty due to different center processings. The centers start their individual data

processing chain at different processing levels. Some centers have a full processing chain, which means that they start at the raw data level, i.e., processing phase data and orbit data, and subsequently bending angle to temperature data (centers GFZ, JPL, UCAR). Some centers start with their processing only at the level of phase data. They use phase data and orbit data processed by another center as input for their processing of atmospheric variables from bending angle to temperature. The centers DMI and WEGC use phase and orbit data from UCAR/CDAAC in this study. We revised the respective paragraph in order to make this clearer. It now reads: "Three of the RO centers (GFZ, JPL, UCAR) have the full processing chain implemented, going from raw data level to atmospheric variables. Two centers (DMI and WEGC) start their processing at phase data level in this study, by using phase data and orbit data from UCAR/CDAAC (COSMIC Data Analysis and Archive Center). Thus, as some centers start with the same phase and orbit data (from UCAR), the products from raw data to atmospheric parameters are not strictly independent for these centers."

*Comment 5

The subscript s refers to the satellite receiver (not transmitter). It may be useful to indicate 'receiver'.

*Response 5

The subscript refers to the satellite mission (not to a single receiver). In the revised manuscript, we denote it as "satellite mission (s)".

*Comment 6

Were equations (1) to (6) applied to the subset of common profiles processed by all centers?

*Response 6

Yes, we added a sentence on this in section 3 of the revised manuscript text at the beginning of paragraph 3: "The common subset of data was analyzed further."

*Comment 7

"only JPL provided a smaller amount": Looking at figure 1, one sees that JPL did provide a smaller amount indeed, of about 10,000 profiles per month, compared to the pack of other producers. However, quite interestingly, the common subset of profiles, between all producers, is also 10,000 lower than the JPL count. This near-match in the differences (from other producers to JPL, and then from JPL to common subset) is quite puzzling. Could it come from an unexpected issue in individual ID assignment (e.g., a shift by a minute or so), which would make many JPL profiles not match the other IDs?

*Response 7

The number of profiles plotted in Fig.1 is the number delivered by each center for this study. As noted in the manuscript, quality control is handled differently at each center, which means that not always the same set of profiles is delivered by each center. The profiles were collocated based on a unique event identifier (ID) including information on receiver ID, GPS satellite ID, date, and time of the observation. The JPL profiles are matched correctly to the other IDs. The fact that JPL produces less COSMIC data is likely due to a combination of retrieval processing loss and some data not fully processed.

*Comment 8

"we are interested in the structural uncertainty of trends represented by the standard deviation of the n_center individual center trends": Unless I misunderstood something, given that n_centers is (at most) 5, this means that the central metric of the paper is a standard deviation based on a population of 5 members. How reliable is a standard deviation based on so few members? This needs to be discussed. In the future, wouldn't there be a more robust estimate that can computed, to characterize this spread, or inconsistencies, given a such small sample?

*Response 8

We investigate the consistency of RO time series. We assess if final data products from different processing systems (based on the same raw measurements) are consistent and if they give consistent trend information. Therefore, we estimate the structural uncertainty of RO time series and trends. As a measure for structural uncertainty, we analyze the spread in difference time series and the spread in trends (see Wigley, 2006). Analyzing difference time series removes the variability that is common to the data sets and isolates those differences that may be due to differences in data set production methods, i.e., structural uncertainty of the data records (Fig. 4 to 7). Correspondingly, we chose the spread of trends (the standard deviation) as a measure of the structural uncertainty in trends (Fig. 8 to 11). As discussed in our response to comment 2 above, the challenge is to quantify the true spread given a limited number of data sets. Thus, it is important to have several independent data records, but at least a minimum of three independently derived datasets are regarded necessary for an estimate of the structural uncertainty (Thorne, 2005). In section 3 of the revised manuscript, we state this now more explicitly: "The spread of the anomaly difference trends and the spread of the center trends were used for estimating the structural uncertainty (Wigley, 2006) of RO records. For each atmospheric variable and satellite mission, we computed the linear trend over the respective time period for the all-center mean, and for each center. The standard deviation of the center trends was finally used as a measure of the spread."

*Comment 9

The fact that the spread in physical temperatures is reduced so much, from the spread in dry temperatures, needs to be explained. Does it point to the fact that the products use similar (background) constraints in the retrieval, and then correspondingly that all products are probably quite representative of these constraints?

*Response 9

Actually, the spread in physical temperature is similar to the spread in dry temperature or even a bit larger for physical temperature (see Fig. 12). As explained in section 2 and in section 4.1, the different moist air retrieval implementations include a priori information, which introduces uncertainties in physical temperature. Throughout the manuscript text, we state that the spread in physical temperature is similar to that in dry temperature, except for one sentence at the end of section 4.3. We revised that sentence accordingly. It now reads: "Physical temperature shows similar uncertainty at lower altitudes, however, above about 30 km it can be larger than for dry temperature due to a priori information in moist air retrievals."

*Comment 10

The difference found at high latitudes is one very interesting result of this paper. This is mentioned as being related to Arctic SSW (something which correlates well by looking at the 60N-90N timeseries and the occurrences of peaks in winter). I would think this deserves a separate sub-section in discussion, with additional results to go a bit further. Is it possible to illustrate the influence of the different strategies for high-altitude initialization in these situations, e.g., by picking a particular SSW event, and showing individual profiles?

*Response 10

Thank you for addressing this point. The main objective of this work is on the long-term stability of RO products, and to inform about strengths and limitations of the products. Analyzing the center-specific potentials for improvement are within the responsibility of the respective centers. RO processing systems are under continuous development and are therefore expected to improve. This work shall support these efforts but further detailed analyses are out of scope here.

*Comment 11

Throughout the paper, all statements making the link to the GCOS stability requirement

need to be revised, as they all fail to include the other sources of uncertainty affecting stability (other than differences in processing).

*Response 11

We disagree on this point as GNSS RO measurements are regarded as benchmark observations (Goody et al. 1998; GCOS 2011). Benchmark measurements are defined as measurements that can be related to absolute standards and may be repeated at any subsequent time with precise comparability (Goody et al., 1998; 2002), such that the accuracy of the record archived today can be verified by future generations. Traceability to standards of the international system of units (SI) is the foundation for benchmark measurements (e.g., Leroy et al., 2006; Ohring, 2007). The GNSS RO technique is such a stable measurement technique because it is based on highly precise and stable atomic clocks, which ensure long-term frequency stability (e.g., Steiner et al. 2011). The fundamental measurement is the GNSS signal phase change as function of time, measured with great precision, solely in terms of timing, and is an absolute measurement and SI-traceable (e.g., Ohring, 2007). Therefore, separate data sets from different RO missions can be combined without intercalibration to a give long-term stable fundamental climate data record (FCDR). RO bending angle is given as one example of a FCDR by GCOS (2016). All characteristics of RO are described in the introduction section of the manuscript. We revised paragraph 3 and included a definition of long-term stability and a further reference. The paragraph now reads: "For climate observations, the accuracy requirement is much more stringent than for weather observations (Trenberth et al., 2013). However, the key attribute is long-term stability, defined as the extent to which uncertainty of measurement remains constant with time (GCOS, 2016). The uncertainty of the measurement must be smaller than the signal expected for decadal change (Ohring et al., 2005; Bojinski et al., 2014). Accordingly, ECV product requirements for air temperature include global coverage, a vertical resolution of 1–2 km in the troposphere and the stratosphere, a horizontal resolution of 100 km, a measurement uncertainty of 0.5 K, and a stability of 0.05 K per decade

(GCOS, 2016). For a definition of the metrological quantities we refer to Annex B of GCOS (2016) and to JCGM (2012)."

*Comment 12

Typo: 'on exemplary' -> one exemplary

*Response 12

Corrected.

*Comment 13

Table 1, impact height is only defined for UCAR. Do the other centers use different definitions? Shouldn't this have been the same definition for all? In other terms, isn't there a RO community-approved definition of 'impact height'?

*Response 13

There is a consistent common definition of vertical geolocation for RO parameters (but a definition was only included in the UCAR description in Table 1). We revised Table 1. We changed the description for UCAR to have it consistent with the description for the other centers. The revised sentence reads: "JGM2 geoid undulation is used to calculate MSL altitude." We included the common definition for impact altitude used by all centers, which reads: "Bending angle is given as a function of impact altitude, i.e. impact parameter minus radius of curvature minus the undulation of the geoid. The impact parameter is defined as the perpendicular distance between the local center of curvature and the ray path from the GPS satellite."

*Comment 14

Table 1 indicates several vertical reference frames, not always WGS-84 ellipsoid and EGM-96 geoid. As a reminder, WMO Executive Council 59 (in 2007) adopted a draft resolution proposed by the Commission for Basic Systems of these two elements (WGS-84 and EGM-96) as the fundamental bases for vertically referencing all station

observations. This choice was also relayed in the coordinated satellite community for geostationary products, by CGMS in 2011 ("LRIT/HRIT Global Specification"). It is not so much a matter of choosing the 'best' reference frame for each observing system, but one that is fit for purpose and a unique standard in a community, so as to avoid introducing artificial discrepancies/differences between products. Using different reference frames can only create artificial differences. It could hence be useful to make a note that some of the differences between data producers presented in Table 1, such as this one, will eventually need resolving.

\*Response 14

Thank you for raising this point. We addressed the topic of vertical geolocation and related uncertainties in the work of Scherllin-Pirscher et al. (2017). They investigated the implication of different geoid choices, e.g., between EGM-96 and JGM-3, on geopotential height. Differences between the results from these two models were found smaller than $1-2$ m up to 35 km geopotential height, which is considered very small to negligible. For further details we refer to Scherllin-Pirscher et al. (2017).

\*Comment 15

Figure 1, it is unclear why the number of points differ by altitude, even though the list of profiles is supposedly common for all centers. Is this caused by different QC at each vertical level?

\*Response 15

We assume that this comment refers to Fig. 2. This is not caused by different QC, which is assigned per profile. The different number of points can occur if for one center invalid values appear at certain altitude levels.

\*Comment 16

Figures 4 to 7, the equations (shown in the legend inside each plot) are too small to be legible; they could go into a new table, or, better yet, be summarized in a graphic, in a

similar form as Fig. 12.

*Response 16

We revised Fig. 4 to 7 and enlarged the fonts. We improved and enlarged the legends. We removed the inserts in the subpanels except for the global differences as the values of difference trends are described in the manuscript text anyway.

*Comment 17

Figure 8 to 11 pack, in total, over 700 vertical profiles. Surely, there must be a way to summarize this into a manageable amount of information, for readers to grasp the message. These plots are surely of value in a supplement, though.

*Response 17

Figure 12 actually summarizes the final information from Fig. 8 to 11 in a condensed format, but Fig. 12 does not give the full vertical information. Fig. 8 to 11 summarize relevant information at the full vertical resolution (one of the main characteristics of RO) for 6 parameters, 4 satellite missions, 5 centers, 6 latitude zones, and up to 600 altitude levels. The figures graphically convey (i) differences between processing centers, (ii) that uncertainties in trends becoming larger at higher altitudes, (iii) that uncertainties propagate to lower altitudes for more derived parameters in the processing chain, (iv) that uncertainties are smaller at higher altitudes for the RO missions GRACE and METOP. We think that we convey all relevant information in an effective way for a comprehensive and overall picture on the consistency of RO data sets.

**References:

GCOS: Systematic observation requirements for satellite-based data products for climate, GCOS-154, World Meteorological Organization. [online] Available from: https://library.wmo.int/doc_num.php?explnum_id=3710 (Accessed 3 September 2019), 2011.

GCOS: The global observing system for climate: implementation needs, GCOS-200, World Meteorological Organization. [online] Available from: https://library.wmo.int/doc_num.php?explnum_id=3417 (Accessed 3 September 2019), 2016.

Goody, R., Anderson, J., Karl, T., Balstad Miller, R., North, G., Simpson, J., Stephens, G. and Washington, W.: Why We Should Monitor the Climate, Bull. Amer. Meteor. Soc, 83(6), 873–878, doi:10.1175/1520-0477(2002)083<0873:WWSMTC>2.3.CO;2, 2002.

Goody, R., Anderson, J. and North, G.: Testing Climate Models: An Approach, Bull. Amer. Meteor. Soc., 79(11), 2541–2549, doi:10.1175/1520-0477(1998)079<2541:TCMAA>2.0.CO;2, 1998.

JCGM: International vocabulary of metrology—Basic and general concepts and associated terms (VIM 3rd edition), Tech. Rep. JCGM 200:2012, Joint Committee for Guides in Metrology, Office BIPM, Paris. [online] Available from: https://www.bipm.org/utils/common/documents/jcgm/JCGM_200_2012.pdf, 2012.

Leroy, S. S., Anderson, J. G. and Dykema, J. A.: Climate benchmarking using GNSS occultation, in Atmosphere and Climate: Studies by Occultation Methods, edited by U. Foelsche, G. Kirchengast, and A. Steiner, pp. 287–301, Springer-Verlag Berlin Heidelberg., 2006.

Ohring, G. (Ed): Achieving satellite instrument calibration for climate change, ASIC3 Workshop Report, NOAA/NESDIS., 2007.

Scherllin-Pirscher, B., Steiner, A. K., Kirchengast, G., Schwärz, M. and Leroy, S. S.: The power of vertical geolocation of atmospheric profiles from GNSS radio occultation, J. Geophys. Res. Atmos., 2016JD025902, doi:10.1002/2016JD025902, 2017.

Thorne, P. W.: Revisiting radiosonde upper air temperatures from 1958 to 2002, J. Geophys. Res, 110(D18), doi:10.1029/2004JD005753, 2005.

Wigley, T. M. L.: Statistical Issues Regarding Trends, in Temperature Trends in the

Lower Atmosphere: Steps for Understanding and Reconciling Differences, edited by T. R. Karl, S. J. Hassol, C. D. Miller, and W. L. Murray, Washington, D. C., 2006.
* * *

---

## Author Comment (AC2) · 24 Feb 2020

Many thanks to the reviewer for the positive reception of our paper and for the constructive comments and questions. We will revise the manuscript accordingly. Please find our responses to all comments below.

**Specific comments

Comment 1.

Page 2, Line 2: Suggest inserting "in situ" before obserservations, i.e.  => " in situ observations" because it could be argued that there are many satellite radiances available.

[Figure]

Response 1.

We prefer to keep this formulation as it is a general statement and not specific to in-situ observations.

Comment 2.

Page 2, Line 15: the accuracy requirements, 0.1 K for climate and 1 K for NWP, need further explanation. The 1 K for NWP is presumably a random error for a given observation, but what is the definition of the 0.1 K requirement? Further, on Page 3, line 25 it says the observational uncertainty estimate for an individual RO observation is 0.7 K near the tropopause. Is this consistent with the 0.1 K climate requirement? Please clarify. Similarly, clarify "measurement uncertainty of 0.5 K" on Page 2, line 19.

Response 2.

The accuracy requirement numbers in line 15 were "just" given as an example by Trenberth et al. (2013) to explain that accuracy for climate observations needs to be more stringent than for weather observations as changes in temperature over decades are small compared to daily variability. The established numbers for climate monitoring are defined in GCOS (2016), which we cite in the same paragraph. To avoid confusion, we removed the numbers from line 15. The observational uncertainty stated on page 3, line 25, is given for individual RO profiles, and not for monthly averaged climate fields. We further clarified the metrological terms in the revised manuscript on page 2, lines 15–21 and included the following reference. Reference: JCGM (2012), International vocabulary of metrology—Basic and general concepts and associated terms (VIM 3rd edition), Tech. Rep. JCGM 200:2012, Joint Committee for Guides in Metrology, Office BIPM, Paris.

Comment 3.

Page 5, Line 18: "Two coherent carrier signals ...". This sentence may give the impression that the ionospheric correction is in phase space. Please clarify.
Response 3.

We rephrased this sentence to: "Two coherent carrier signals are transmitted, in case of the U.S. Global Positioning System (GPS) at wavelengths of 0.19 m (L1 signal) and 0.24 m (L2 signal) (Hofmann-Wellenhof et al., 2008; Teunissen and Montenbruck, 2017), which enables removing contributions due to Earth's ionosphere in a later retrieval step. "

Comment 4.

Page 6, Line 2: It probably should be noted that no centre is currently trying to correct residual ionospheric errors using, for example, techniques such as those in Danzer etal (2015). Although there is still work required to demonstrate this approach (Danzer etal, 2019 submitted), it should be noted that residual ionospheric errors are a potentially a common error at all the centres.

Response 4.

We thank the reviewer for pointing to this. We included the suggested references and added a sentence on residual ionospheric error on page 6, first paragraph (line 3–4): "Current research aims at further minimization of the residual ionospheric error (Danzer et al. 2015)." Reference: Danzer, J., Healy, S. B. and Culverwell, I. D.: A simulation study with a new residual ionospheric error model for GPS radio occultation climatologies, Atmospheric Measurement Techniques, 8(8), 3395–3404, doi:10.5194/amt-8-3395-2015, 2015.

Comment 5.

Page 6, Line 9: Some NWP centres have moved away from Smith and Weintraub (1953) to potentially more accurate formulations including both updates to the assumed C02 concentration and non-ideal gas effects. This is mainly as a result of work by Dr Aparicio. See Appicio and Larosche (2011) and references therein, Cucurull et al (2013), Healy (2011). The NWP implementations should be noted.

Response 5.

We updated the respective sentence and included all suggested references: "... and is given by the Smith Weintraub formula (Smith and Weintraub, 1953) or updated formulations (Aparicio and Laroche, 2011; Healy, 2011; Cucurull et al., 2013)."

Comment 6.

Figure 2: The Metop bending angles for WEGC at _15 km seem to be an outlier. Any reason for this?

Response 6.

We thank the reviewer for pointing to this. We can confirm that some outliers are the reason. For a better statistical representation, we consistently recomputed all monthly statistics using the median. We included the revised plots in the manuscript.

Comment 7.

Figure 3: The JPL and UCAR appear to have almost equal and opposite bending angle biases. Please discuss.

Response 7.

It appears that JPL and UCAR have almost equal and opposite bending angle biases, because the centers are plotted with respect to the all-center mean. If one center has a larger deviation, this is counter-balanced by the other centers. This is a limitation of the comparison to the all-center mean. A better identification of which data sets have a larger deviation than others is possible in Fig. 9 (top) for trends. In the concrete case, the JPL bending angle deviates more whereas the other centers show more overlap. We added a sentence in the revised manuscript text to emphasize that the plots are with respect to the all-center mean, in section 4.1 at the end of the first paragraph: "Note that deviations of one center are counter-balanced by other centers due to referencing to the all-center mean."

[Figure]

Comment 8.

Page 10, line 21: "Above this altitude, WEGC ...". It might be worth adding that the WEGC dry-temp and temperature differences above 16 km shown in Figure 3 are because of different all centre mean values.

Response 8.

We added the following explanation in the revised version of the manuscript text : "Above this altitude, WEGC dry and physical temperature are the same. However, in Fig. 3, differences are shown with respect to the all-center mean, and the latter is different for dry and physical temperature."

Comment 9.

Section 4.2, Page 11, Lines 9-10. "Larger variability ..." for JPL is likely due to bending angle extrapolation? Why is extrapolation relevant here?

Response 9.

Actually it is not relevant here because the "raw" bending angle is shown in Fig. 4a. We therefore removed this part of the sentence in the revised text.

Comment 10.

Page 14, line 16. When quoting the uncertainty in the trends , e.g. "0.06 %", include "per decade".

Response 10.

We included "per decade" throughout the revised manuscript text.

Comment 11.

Page 14, line 26. The bending angles are found to be consistent up to 50 km because they are less sensitive to a priori information. Ringer and Healy (2006) suggested monitoring the climate in bending angle space for this reason, although the interpretation

of bending angle trends is more complicated. Consider adding this reference.

Response 11.

According to the reviewer's suggestion, we added the following statement in the revised manuscript text in the last paragraph of section 4: "Bending angles are found to be consistent up to 50 km because they are less sensitive to a priori information and thus useful for climate monitoring (Ringer and Healy, 2008)."

**Technical suggestions

Comment 12.

The text on many figures is still very difficult to read. Figure 4b, 5b, 6b, 7b, 8-18km dry temperature time series. The vertical ranges/axes could be expanded. Figure 6a, 7a. Better vertical ranges could be used in these figures.

Response 12.

We revised Figures 4 to 7 and enlarged the fonts. We removed the inserts in the subpanels except for the global differences. The difference trends are described in the manuscript text anyway. Regarding changing the vertical axes for some sub-panels, we decided to keep the range of the vertical axes the same for all panels. The main purpose of the differences time series plots is to give an overview on the consistency of the data for different altitude layers and over all satellite missions. Keeping the same axis ranges makes it better comparable and gives the reader and data users an overview on the performance and accuracy of the different RO missions. It is not about zooming into the details in each difference time series but conveying the big picture.

Comment 13. Suggested References

Cucurull, L., Derber, J. C., and Purser, R. J. (2013), A bending angle forward operator for global positioning system radio occultation measurements, J. Geophys. Res. Atmos., 118, 14-28, doi:10.1029/2012JD017782.

Aparicio, J. M., and Laroche, S. (2011), An evaluation of the expression of the atmospheric refractivity for GPS signals, J. Geophys. Res., 116, D11104, doi:10.1029/2010JD015214.

Healy, S. B. ( 2011), Refractivity coefficients used in the assimilation of GPS radio occultation measurements, J. Geophys. Res., 116, D01106, doi:10.1029/2010JD014013.

Ringer, M. A., and Healy, S. B. (2008), Monitoring twenty-first century climate using GPS radio occultation bending angles, Geophys. Res. Lett., 35, L05708, doi:10.1029/2007GL032462.

Response 13.

We included all suggested references.
* * *

---

## Author Response (AR2)

**Manuscript number amt-2019-358**
**Consistency and structural uncertainty of multi-mission GPS radio occultation records**
**by A.K. Steiner et al.**

**Dear Editor,**

in this second revision, we revised the manuscript along the remaining four minor comments of reviewer#2. In addition, we made minor corrections in Fig.8 to 11 by changing lines from thin to bold as stated in the figure captions. Please find in the following a list of changes to the manuscript.

**List of minor changes**

- We revised Table 1 and included specified information on the use of GPS orbits (along comment 1 of the reviewer). We made some further minor clarifications in Table 1.
- We made some minor changes in Table 2 to make it more clear, i.e. we moved the information about the common data set used in this study from the bottom to the top.
- We rephrased the last sentence in section 4 (along comment 3 of the reviewer).
- We made minor corrections in Fig.8 to 11. We changed the lines representing the all-center mean trend from thin lines to bold lines. It is now consistent with the description in the figure captions.

We thank you very much for handling our paper.

Kind regards,
Andrea Steiner
on behalf of the authors

**Manuscript number amt-2019-358**
**Consistency and structural uncertainty of multi-mission GPS radio occultation records**
**by A.K. Steiner et al.**

**Response to Referee #2**
We thank the reviewer for a second review of our manuscript and for the remaining minor comments. We revised the manuscript accordingly. Please find our responses below.

**Minor comments:**
**Comment 1**
If any information from IGS is used in the processing, this may be worth referencing/acknowledging.
**Response 1**
We included and specified information on the use of GPS orbit information in the different processings of the centers in the revised Table 1 of the manuscript.

**Comment 2**
Is the smallest structural uncertainty (found for one mission) the result of receiver design?
**Response 2**
This is considered due to advanced receivers (better onboard clocks). We also note that for the structural uncertainty estimate only three centers delivered data for the GRACE and Metop missions, while five centers provided data for CHAMP and four centers provided data for F3C.
We discuss this already in the manuscript at two places:
Section 4.2 last paragraph:
*"Comparing results of the four RO missions, we find the highest consistency for GRACE and Metop between the centers. CHAMP and F3C show a bit larger differences, above 25 km (CHAMP) and at high latitudes (F3C). Apart from small features, the results are very consistent at 8–30 km. One potential reason for the higher consistency of GRACE and Metop records is considered to be technological advances on the newer satellite generations. Partly it might also be due to that only three centers delivered data for these missions, while five centers provided data for CHAMP and four centers provided data for F3C."*
Section 5, paragraph 4: *"… data products from the newer satellite missions F3C, and specifically GRACE and Metop, are usable to higher altitudes due to advanced receivers (better onboard clocks) and lower bending angle noise at higher altitudes."*

**Comment 3**
End of section 4, are the structural uncertainties the only sources of uncertainties to consider when assessing conformity to the GCOS stability requirements?
**Response 3**
As discussed in our former response to comment 11 of referee#2, RO is considered a stable measurement technique as it is based on highly precise and stable atomic clocks. Therefore structural uncertainty/different processing choices encompass the main source of uncertainty in the case of RO. For obtaining a reasonable estimate of this uncertainty, a sufficient diversity of the basic data processing (from the raw data to orbit and excess phase data level) as well as of retrieval processing (from excess phase level to finally-retrieved atmospheric profiles) need be available. We account for this by a clear quality control of the input data, in order to statistically safeguard the basic characteristics of precision and long-term stability of the raw data that flow into the processings, and by the diversity of processing setups and implementation variants of the five different centers that we comprehensively summarize in Table 1.

We rephrased the last sentence of section 4 to:

[revised manuscript text omitted]